# Ceapins are a new class of unfolded protein response inhibitors, selectively targeting the ATF6α branch

Ciara M Gallagher[1]*, Carolina Garri[1,2], Erica L Cain[1], Kenny Kean-Hooi Ang[3], Christopher G Wilson[3], Steven Chen[3], Brian R Hearn[3], Priyadarshini Jaishankar[3], Andres Aranda-Diaz[1], Michelle R Arkin[3], Adam R Renslo[3], Peter Walter[1]*

[1]Department of Biochemistry and Biophysics, Howard Hughes Medical Institute, University of California, San Francisco, United States; [2]Fundación Ciencia Para la Vida, Santiago, Chile; [3]Small Molecule Discovery Center, Department of Pharmaceutical Chemistry, University of California, San Francisco, San Francisco, United States

**Abstract** The membrane-bound transcription factor ATF6α plays a cytoprotective role in the unfolded protein response (UPR), required for cells to survive ER stress. Activation of ATF6α promotes cell survival in cancer models. We used cell-based screens to discover and develop Ceapins, a class of pyrazole amides, that block ATF6α signaling in response to ER stress. Ceapins sensitize cells to ER stress without impacting viability of unstressed cells. Ceapins are highly specific inhibitors of ATF6α signaling, not affecting signaling through the other branches of the UPR, or proteolytic processing of its close homolog ATF6β or SREBP (a cholesterol-regulated transcription factor), both activated by the same proteases. Ceapins are first-in-class inhibitors that can be used to explore both the mechanism of activation of ATF6α and its role in pathological settings. The discovery of Ceapins now enables pharmacological modulation all three UPR branches either singly or in combination.

*For correspondence: ciara@ walterlab.ucsf.edu (CMG); peter@ walterlab.ucsf.edu (PW)

**Competing interests:** The authors declare that no competing interests exist.

## Introduction

Most secreted and transmembrane proteins utilize the endoplasmic reticulum (ER) as a dedicated folding compartment. It is estimated that about one third of all newly synthesized proteins pass through the ER, where they fold and assemble into multi-subunit complexes, and where post-translational modifications such as disulfide bridge formation and glycosylation occur (*Braakman and Hebert, 2013*). Dedicated quality control mechanisms ensure that only properly folded proteins exit the ER. These mechanisms are part of the cell's 'proteostasis network' and include chaperone systems to aid in protein folding and ER associated degradation (ERAD) to remove terminally misfolded proteins (*Ruggiano et al., 2014*). In addition, the ER has the ability to adjust its folding capacity upon demand through activation of a homeostatic signaling network, called the unfolded protein response (UPR). The UPR directs cell fate - cells that cannot restore homeostasis then initiate apoptosis to prevent secretion or cell surface presentation of misfolded, non-functional proteins (*Lin et al., 2007*; *Lu et al., 2014*).

Three principal transmembrane sensors of the UPR independently monitor folding stress in the ER – IRE1, PERK and ATF6 (*Gardner et al., 2013*). These UPR branches function cooperatively to decrease the load of incoming polypeptides and to increase both the protein folding and degradative capacity of the ER through regulation of transcription and translation.

**eLife digest** Newly made proteins must be folded into specific three-dimensional shapes before they can perform their roles in cells. Many proteins are folded in a cell compartment called the endoplasmic reticulum. The cell closely monitors the quality of the work done by this compartment. If the endoplasmic reticulum has more proteins to fold than it can handle, unfolded or misfolded proteins accumulate and trigger a stress response called the unfolded protein response. This increases the capacity of the endoplasmic reticulum to fold proteins to match the demand. However, if the stress persists, then the unfolded protein response instructs the cell to die to protect the rest of the body.

A protein called ATF6α is one of three branches of the unfolded protein response. This protein is found in the endoplasmic reticulum where it is inactive. Endoplasmic stress causes ATF6α to move from the endoplasmic reticulum to another compartment called the Golgi apparatus. There, two enzymes cut ATF6α to release a fragment of the protein that then moves to the nucleus to increase the production of the machinery needed to fold proteins in the endoplasmic reticulum.

Errors in protein folding can cause serious diseases in humans and other animals. Drugs that target ATF6α might be able to regulate part of the unfolded protein response to treat these diseases. However, no drugs that act on ATF6α had been identified. Now, two groups of researchers have independently identified small molecules that specifically target ATF6α.

Gallagher et al. screened over 100,000 compounds for their ability to reduce the activity of ATF6α-regulated genes. The experiments reveal that a class of small molecules termed Ceapins can selectively block the activity of ATF6α during endoplasmic reticulum stress, but had no effect on other proteins involved in the unfolded protein response. Furthermore, when human cells experiencing stress were treated with Ceapins, a greater number of cells died in comparison to cells that had not received Ceapins. An accompanying study by Gallagher and Walter reports on the mechanism by which Ceapins act on ATF6α.

Independently, Plate et al. identified a type of small molecule that can activate ATF6. Together, the findings of Gallagher et al. and Plate et al. may lead to the development of new drugs for treating diseases associated with incorrect protein folding in the endoplasmic reticulum.

ATF6 and IRE1 increase the folding capacity of the ER by upregulating transcription of UPR target genes. ATF6 is a membrane-tethered transcription factor activated by regulated trafficking and proteolysis producing ATF6-N, the ATF6 fragment that constitutes the functional transcription factor (*Haze et al., 1999*; *Ye et al., 2000*; *Chen et al., 2002*). In contrast, the highly conserved kinase-endoribonuclease IRE1 removes an intron from the mRNA encoding the UPR effector XBP1 allowing translation of XBP1s ('s' for spliced), the functional transcription factor variant of this protein (*Yoshida et al., 2001*). ATF6-N and XBP1s bind to ER stress response (ERSE) (*Yoshida et al., 1998*; *2000*; *Roy and Lee, 1999*) and unfolded protein response (UPRE) elements (*Yamamoto et al., 2004*), respectively in the promoters of UPR target genes. ATF6 upregulates transcription of chaperones, foldases and lipid synthesis genes (*Wu et al., 2007*; *Yamamoto et al., 2007*; *Adachi et al., 2008*), while XBP1 upregulates ER chaperones and the ERAD machinery (*Lee et al., 2003*; *Acosta-Alvear et al., 2007*). Decreasing the load of proteins entering the ER is coordinated by IRE1 and PERK. Regulated IRE1-dependent mRNA decay (RIDD) cleaves ER-targeted mRNAs leading to their degradation (*Hollien et al., 2009*). PERK, a second transmembrane kinase, phosphorylates itself and the α-subunit of the initiation factor (eIF2-α) leading to transient inhibition of cap-dependent translation and an increase in translation of UPR effectors with upstream open reading frames (*Harding et al., 1999*; *Sidrauski et al., 2015*), including the transcription factor ATF4 and the apoptotic effector CHOP. The balance between survival and death is controlled temporally. Both IRE1 and ATF6 signaling attenuate after prolonged ER stress, removing most of the cytoprotective functions of the UPR (*Lin et al., 2007*; *Lu et al., 2014*; *Haze et al., 2001*). PERK signaling is maintained and, through CHOP, commits the cell to apoptosis (*Zinszner et al., 1998*; *Palam et al., 2011*).

This dual capacity of the UPR to boost the protein folding capacity or drive cell death has been implicated in many disease models (*Ryno et al., 2013*). Many small molecule inhibitors and

activators of the PERK and IRE1 enzymes have been isolated (*Maly and Papa, 2014*; *Maurel et al., 2015*; *Mendez et al., 2015*). In contrast, no pharmacological agents promoting the selective modulation of ATF6 have been developed, in part due to the fact that, unlike IRE1 and PERK, ATF6 is not an enzyme.

Two closely related homologs define the ATF6 family of ER stress sensors; ATF6α and ATF6β, which respond to the same stress inducers and are activated with similar kinetics. ATF6α and ATF6β act redundantly during development as single knockout mice are viable and fertile while double knockout animals are pre-implantation lethal (*Yamamoto et al., 2007*). Conversely, ATF6α and ATF6β do not appear to act redundantly during ER stress as ATF6α knockout cells or animals die when challenged with ER stressors (*Wu et al., 2007*; *Yamamoto et al., 2007*). The transcriptional targets of ATF6β and its role during ER stress remain poorly defined.

Regulation of ATF6 signaling is by spatial separation of the substrate, ATF6, in the ER and the proteases, site-1 and site-2 proteases (S1P and S2P, respectively), in the Golgi apparatus: upon ER stress ATF6 moves from the ER to the Golgi apparatus. The mechanism by which ATF6 trafficking is regulated is poorly understood but the transport requires the COPII coat (*Nadanaka et al., 2004*; *Schindler and Schekman, 2009*), which is not unique to ATF6. Inhibition of ATF6 was achieved by inhibiting the proteases that release it from the membrane – S1P and S2P, respectively (*Ye et al., 2000*; *Okada et al., 2003*). These proteases do not uniquely process ATF6 but also play essential roles in regulating cholesterol homeostasis via processing of SREBP (*Brown and Goldstein, 1997*) and lysosome biogenesis via processing of the α/β-subunit precursor of the N-acetylglucosamine-1-phosphotransferase complex (*Marschner et al., 2011*). Their pleiotropic engagement limits the usefulness of S1P and S2P inhibitors for studies of the UPR.

To date, the ATF6 signaling pathway was considered 'undruggable'. Here we developed cell-based screens to identify a series of pyrazole amides as the first selective inhibitors of the ATF6α branch of the UPR. We show in the accompanying manuscript that these compounds trap ATF6α in the ER in discrete foci, which inspired us to name the compounds 'Ceapins' from the Irish verb 'ceap' meaning 'to trap' (*Gallagher and Walter, 2016*). Ceapins do not inhibit activation of either IRE1 or PERK in response to ER stress, nor do they inhibit trafficking and cleavage of SREBP in response to low sterols or ATF6β in response to ER stress. Through structure-activity studies we increased the potency of the series ten-fold to an $IC_{50}$ of 600 nM. Inhibition of ATF6α with Ceapin analogs has no toxicity in unstressed cells but increases the sensitivity of cells to ER stress inducers, closely mimicking the genetic ablation of ATF6α in mice. This makes Ceapins the most selective class of ATF6α inhibitors identified to date and the first to act through a mechanism distinct from protease inhibition or general trafficking between the ER and the Golgi.

## Results

### Isolation of small molecule inhibitors of ATF6 mediated transcription of UPR targets

To isolate small molecule modulators of ATF6 signaling, we used an assay based on the activation of transcription of ATF6 target genes. To this end, we cloned two copies of the ER stress response element (ERSE) upstream of a minimal promoter driving expression of luciferase (*Figure 1A*) into a retroviral vector, which we used to generate a HEK293T-based ERSE-luciferase reporter stable cell line. To induce ER stress, we treated the reporter cells with thapsigargin (Tg), an inhibitor of the ER calcium pump. ER stress causes a $3.8 \pm 0.2$-fold induction of luciferase activity (*Figure 1B*). The ER stress-induced luciferase was not affected by inhibition of IRE1 (*Figure 1—figure supplement 1*) (*Patterson et al., 2011*), indicating that the cell line reported selectively on the ATF6 branch. We screened 106,281 compounds at a single concentration for their ability to inhibit Tg-induced luciferase activity in the reporter cells (*Figure 1C*, see also *Figure 3—figure supplement 2* for overview of screen workflow). About 1% of the compounds (1142) showed >69% inhibition (amounting to three standard deviations from the mean of stressed controls). To focus on ATF6 pathway specific modulators, we next removed from consideration compounds that showed inhibitory activity in analogous assays based on luciferase reporters induced by either IRE1-dependent mRNA splicing (*Figure 1—figure supplement 1*) (*Mendez et al., 2015*) or PERK-dependent translational control (*Sidrauski et al., 2013*). When fresh stocks were retested in dose response assays, all 598 remaining

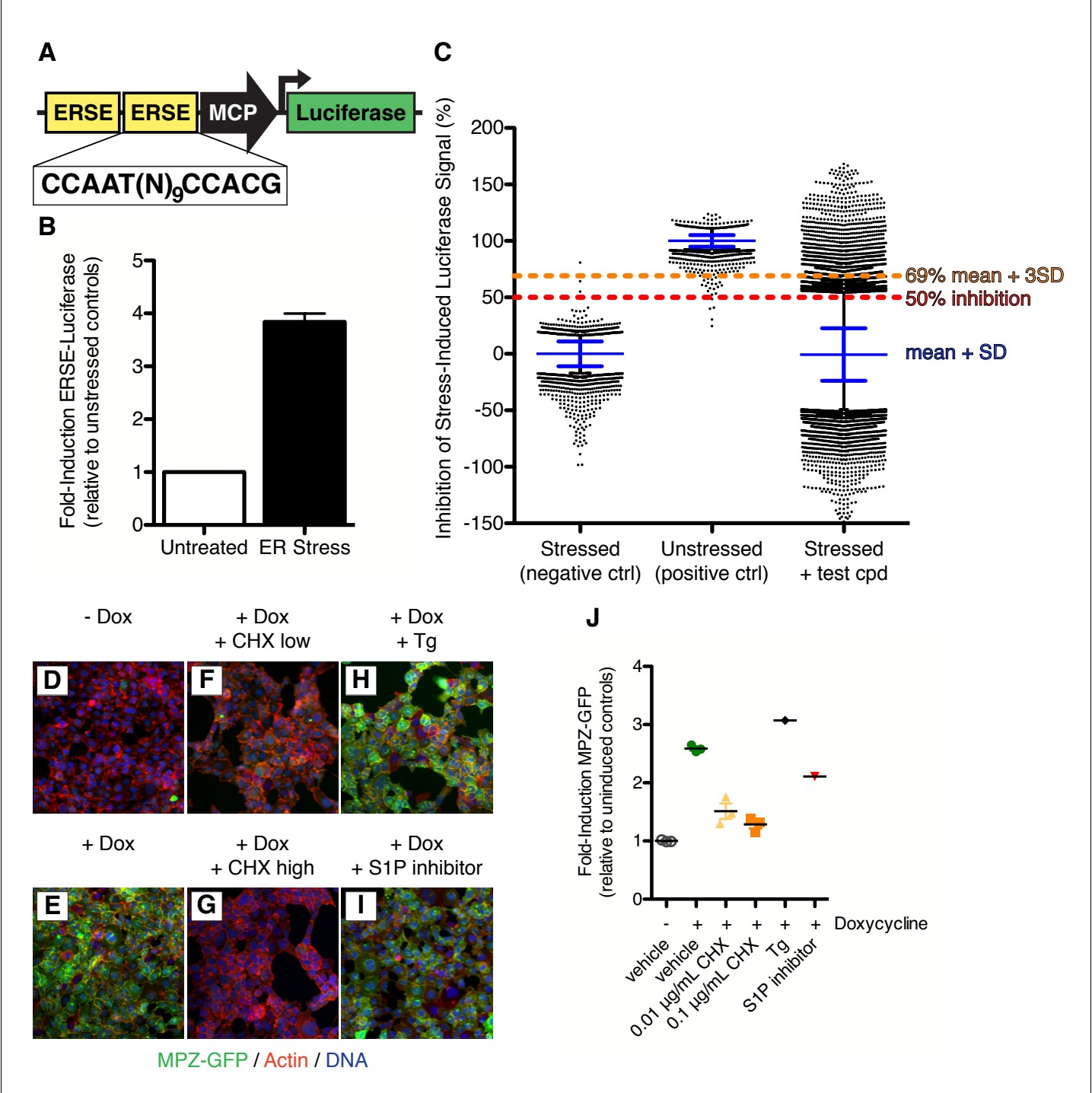

**Figure 1.** Isolation of small molecule inhibitors of ATF6 mediated transcriptional response induced by ER stress. (**A**) Schematic representation of ERSE-luciferase construct used to make screening cell line. Two copies of the ER Stress Response Element (ERSE) were cloned in front of a minimal CMV promoter (MCP) driving expression of luciferase. (**B**) Luciferase activity is induced 3.84 ± 0.16 fold upon ER stress (100 nM Tg) in ERSE-Luciferase 293T reporter cell line. Mean of three independent experiments with at least duplicate wells is plotted; error bars are standard error of the mean. (**C**) Primary screen data from ERSE-luciferase transcriptional reporter cell line. Each plate was internally normalized from 0–100% inhibition using stressed and unstressed controls respectively. 106,281 compounds were added in combination with ER stressor and assayed for their ability to inhibit stress-induced production of ERSE-luciferase. Plot shows % inhibition for each control and compound tested - blue lines denote mean and standard deviation of each population, black dots indicate those wells more than two standard deviations away from the mean of the population.1142 compounds scoring more than three standard deviations from the mean (>69% inhibition, orange line) were classified as hits. (**D–J**). 293 cells expressing doxycycline inducible

*Figure 1 continued on next page*

*Figure 1 continued*

MPZ-GFP were uninduced (**D**) or induced with 50 nM doxycycline without (**E**) or with inhibitors (**F–I**) for seven hours and then fixed and stained for GFP (green), actin (red) and DNA (blue). Inhibitors tested were the protein synthesis inhibitor cycloheximide at either 0.01 µg/mL (**F**) or 0.1 µg/mL (**G**), the ER stressor thapsigargin (100 nM, **H**) or the S1P inhibitor (50 µM Pf-429242, **I**). (**J**) Mean induction of GFP per cell per image was quantified and plotted as fold induction relative to uninduced controls.

The following figure supplement is available for figure 1:

**Figure supplement 1.** IRE1 inhibitor blocks induction of luciferase activity through XBP-luciferase but not ERSE-luciferase.

compounds showed inhibitory activity with an $IC_{50} < 13.5$ µM against Tg-induced luciferase activity in the reporter cells.

We further triaged the reconfirmed compounds to remove non-specific inhibitors of transcription or translation. To this end, we developed a high-throughput assay to determine if the compounds blocked expression of green fluorescent protein-tagged myelin protein zero (MPZ-GFP) under the control of a doxycycline-inducible promoter (*Figure 1D–I*). Addition of doxycycline induced the MPZ-GFP reporter $2.6 \pm 0.06$-fold (*Figure 1E*) compared to uninduced controls (*Figure 1D*). As expected, the translation inhibitor cycloheximide ('CHX', *Figure 1F,G*) prevented MPZ-GFP expression, and treatment of the cells with an ER stressor (*Figure 1H*) or with an S1P inhibitor (*Figure 1I*) (*Hay et al., 2007*; *Hawkins et al., 2008*) did not block doxycycline-induced MPZ-GFP expression. We quantified the fold-induction of MPZ-GFP by doxycycline (*Figure 1J*). Sixty of the 598 compounds isolated in the primary screen inhibited doxycycline-induced MPZ-GFP expression and were removed from further consideration, yielding a collection of 538 compounds that inhibited ER stress-induced ATF6 signaling without inhibiting either transcription or translation or inhibiting signaling through other UPR branches.

## Isolation of small molecule inhibitors of ER stress-induced nuclear translocation of GFP-ATF6α

Each step of ATF6α activation occurs in a different organelle – stress-sensing in the ER, proteolytic processing in the Golgi and activation of transcription in the nucleus. To begin mapping the action of the inhibitors to the steps of ATF6 activation, we next determined the subcellular localization of ATF6α using a cell line that stably expresses GFP-ATF6α (*Figure 2A–D*). Using an antibody against GRP94 to mark the ER and a DNA stain to mark the nucleus, we examined the ratio of GFP intensity between the nucleus and the ER. In unstressed cells, GFP-ATF6α colocalized with GRP94, indicating its predominant localization in the ER (*Figure 2A*). Upon ER stress, GFP-ATF6α translocated to the nucleus and colocalized with DNA (*Figure 2B*, yellow in merged image). As a positive control, when we inhibited ATF6 cleavage by S1P, GFP-ATF6α no longer translocated to the nucleus but accumulated in perinuclear punctae, corresponding to the Golgi apparatus (*Figure 2C*, see also *Figure 3* in *Gallagher and Walter (2016)*). We classified compounds that decreased or inhibited nuclear translocation as 'Class 1 inhibitors'. *Figure 2D* shows Ceapin-A1 as an example in this class showing decreased nuclear GFP signal and perinuclear GFP-ATF6α punctae. We classified compounds that allowed nuclear translocation but inhibited reporter transcription as 'Class 2 inhibitors'. By our definition, Class 2 inhibitors act downstream of ER-Golgi trafficking, proteolysis, and nuclear import of ATF6. These inhibitors may act by preventing DNA binding or interaction with transcriptional co-activators, such as NF-Y (*Yoshida et al., 2000*; *Li et al., 2000*).

To quantify GFP-ATF6α localization, we defined a threshold for activated cells – i.e. cells that responded to ER stress and show nuclear translocation of GFP-ATF6. Using CellProfiler (*Carpenter et al., 2006*), we used the GRP94 and DNA images to generate masks for the ER and nuclei, respectively (*Figure 2—figure supplement 1*). We next calculated the ratio of nuclear to ER GFP signal (nuc:ER ratio) for each cell and plotted the nuc:ER ratios for the unstressed and stressed cells as histograms (*Figure 2E*). The distributions showed a wide range of responses within the population of cells in each well. To compare wells treated with different inhibitors, we extracted from these distributions a single metric representing the percentage of stressed cells for each well, as described in the Methods. This allowed us to convert single cell measurements of GFP-ATF6α from

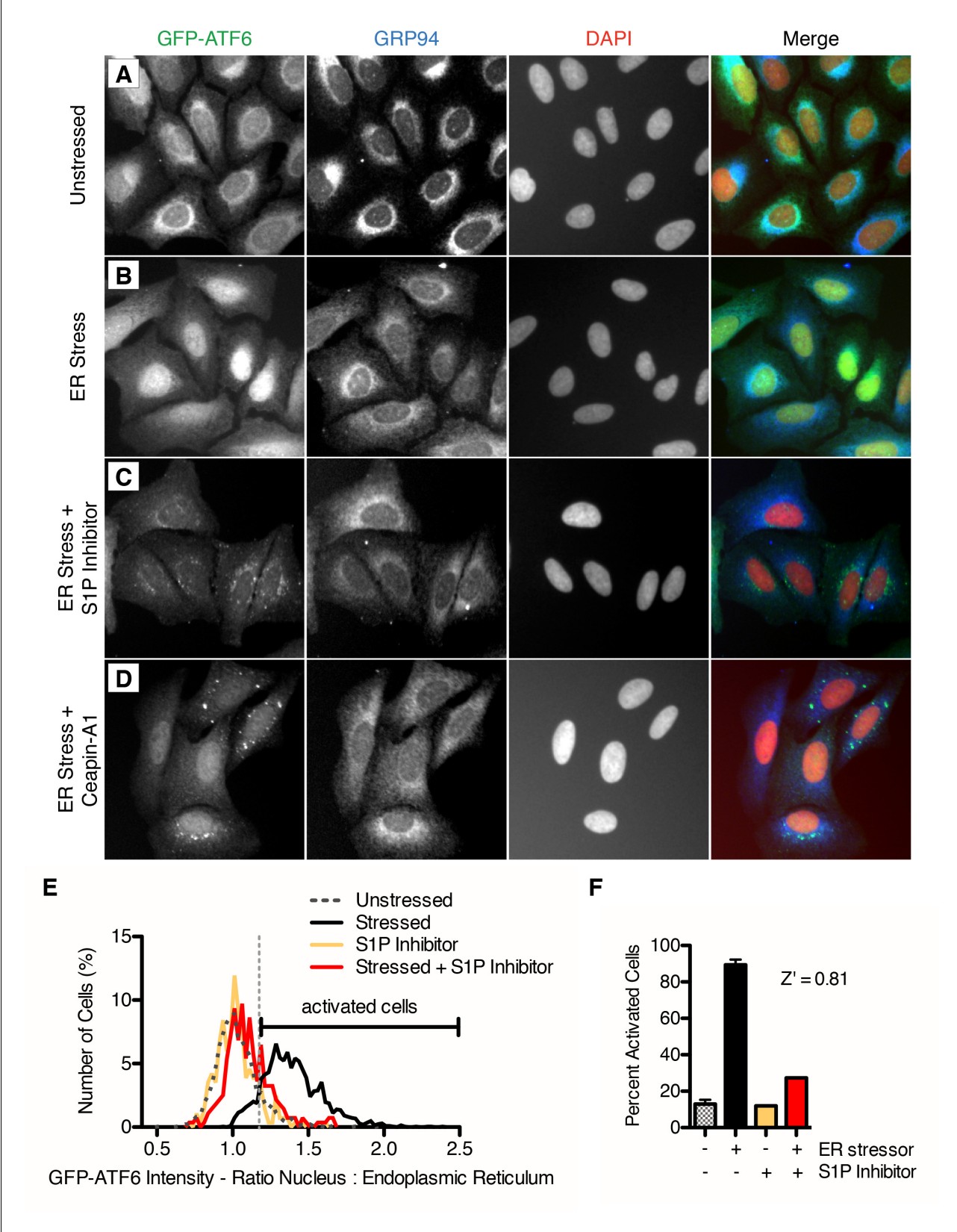

**Figure 2.** Isolation of small molecule inhibitors of ER stress induced nuclear translocation of ATF6. (**A–D**) Nuclear translocation assay in U2-OS GFP-ATF6α cells. U2-OS cells expressing GFP-ATF6α were treated with either vehicle (unstressed, DMSO, A) or ER stressor in the absence (ER stress, 100
*Figure 2 continued on next page*

*Figure 2 continued*

nM Tg, B) or presence of S1P inhibitor (20 μM Pf-429242, C) or screen hit (6.6 μM Ceapin-A1, D). After five hours, cells were fixed and stained for GFP (green), GRP94 (ER marker, blue) and DNA (nuclear marker, DAPI, red). (E–F) Quantification of nuclear translocation assay. (E) Single cell ratios of nuclear: ER GFP intensity were calculated for four images per well for each treatment (unstressed and stressed control wells are present seven times per plate) and plotted as histograms. For each plate, the minimum nuclear: ER ratio where the percentage of stressed cells is greater than the percentage unstressed cells is calculated and annotated as the threshold for activation by ER stress (light grey vertical dashed line). (F) For each plate, the percent activation by ER stress is calculated for the control wells (unstressed n = 1904, ER stress n = 2095, unstressed + S1P inhibitor n = 366, stressed + S1P inhibitor n = 330 cells) and used to generate a Z' score for the plate.

The following figure supplements are available for figure 2:

**Figure supplement 1.** Annotation of nuclear and ER regions used for calculation of the ratio of nuclear to ER GFP-ATF6α signal for each cell.

**Figure supplement 2.** Example of a heat map for a plate from the nuclear translocation assay secondary screen showing percent inhibition of test compounds compared to controls.

**Figure supplement 3.** Ceapin-A1 inhibits nuclear translocation of GFP-ATF6.

**Figure supplement 4.** Combining data from high content assays identified non-specific inhibitors of trafficking and toxic compounds.

the images to compare across plates. We performed these analyses in biological triplicates. Statistical evaluation of unstressed and stressed controls yielded a mean Z' of 0.7 +/- 0.18, which is an exceptionally robust read-out for a cell- and image-based high-throughput assay. We further validated the analyses using S1P inhibitor as a positive control (*Figure 2E and F*). An example of a heat map for one plate assessing the inhibitory activity of different compounds is shown in *Figure 2—figure supplement 2*. An example of a hit from this assay, Ceapin-A1 is shown in *Figure 2—figure supplement 3*. Analysis of the data using t-tests instead of the threshold method gave the same results, indicating that thresholding did not introduce a bias into the data analysis. Of 598 compounds tested in this way, 85 showed robust inhibition of ER stress-induced nuclear translocation of GFP-ATF6α above three standard deviations from the mean of the negative (Tg-alone) control.

We further triaged top-scoring compounds to remove false positives. Toxic compounds scored as hits in the ERSE-luciferase assay but identified in image-based assays as compounds that reduced the cell number per well below 50% of stressed controls (e.g., wells E5, G5 and G10 in *Figure 2—figure supplement 2*; sample images in *Figure 2—figure supplement 4E and J*). They were removed from further consideration. Likewise, we removed 18 compounds that inhibited nuclear translocation of GFP-ATF6α by globally inhibiting protein trafficking from the ER by examining the image data from our MPZ-GFP assay (*Figure 2—figure supplement 4C,D, H and I*). MPZ-GFP is targeted to the ER where it is folded prior to export to the plasma membrane (*Figure 2—figure supplement 4G*) (*Pennuto et al., 2008*). Furthermore, we applied a potency cut-off of $IC_{50} < 5$ μM to Class 2 (transcriptional) inhibitors. We next analyzed the chemical structures of the compounds to remove pan-assay interference compounds (PAINS) (*Baell and Holloway, 2010*; *Dahlin et al., 2015*). After completion of these assays, we were left with 38 Class 1 and 128 Class 2 inhibitors, of which 29 and 70 were repurchased.

## Ceapins, a class of pyrazole amides that inhibit ATF6 but not SREBP processing

We performed dose-response ERSE-luciferase assays on the repurchased compounds using two different ER stressors – Tg or tunicamycin (Tm). Tm inhibits N-linked glycosylation and induces the UPR. Of 99 repurchased compounds, 98 showed inhibitory activity with Tg, of which 82 also inhibited tunicamycin induced ERSE-luciferase. We next assessed compounds displaying well-shaped dose-response curves and $IC_{50}$'s < 19 μM in both Tg and Tm ERSE-luciferase assays for their ability to inhibit induction of endogenous ATF6α target genes, GRP78 (encoding the ER HSP70 BiP) and HERPUD1 (encoding a ubiquitin-domain-containing protein involved in ERAD) (*Figure 3A*, and *Figure 3—figure supplement 1*) (*Wu et al., 2007*; *Yamamoto et al., 2007*; *Adachi et al., 2008*). This qPCR assay did not rely on the presence of reporters and proved the most stringent (a workflow for

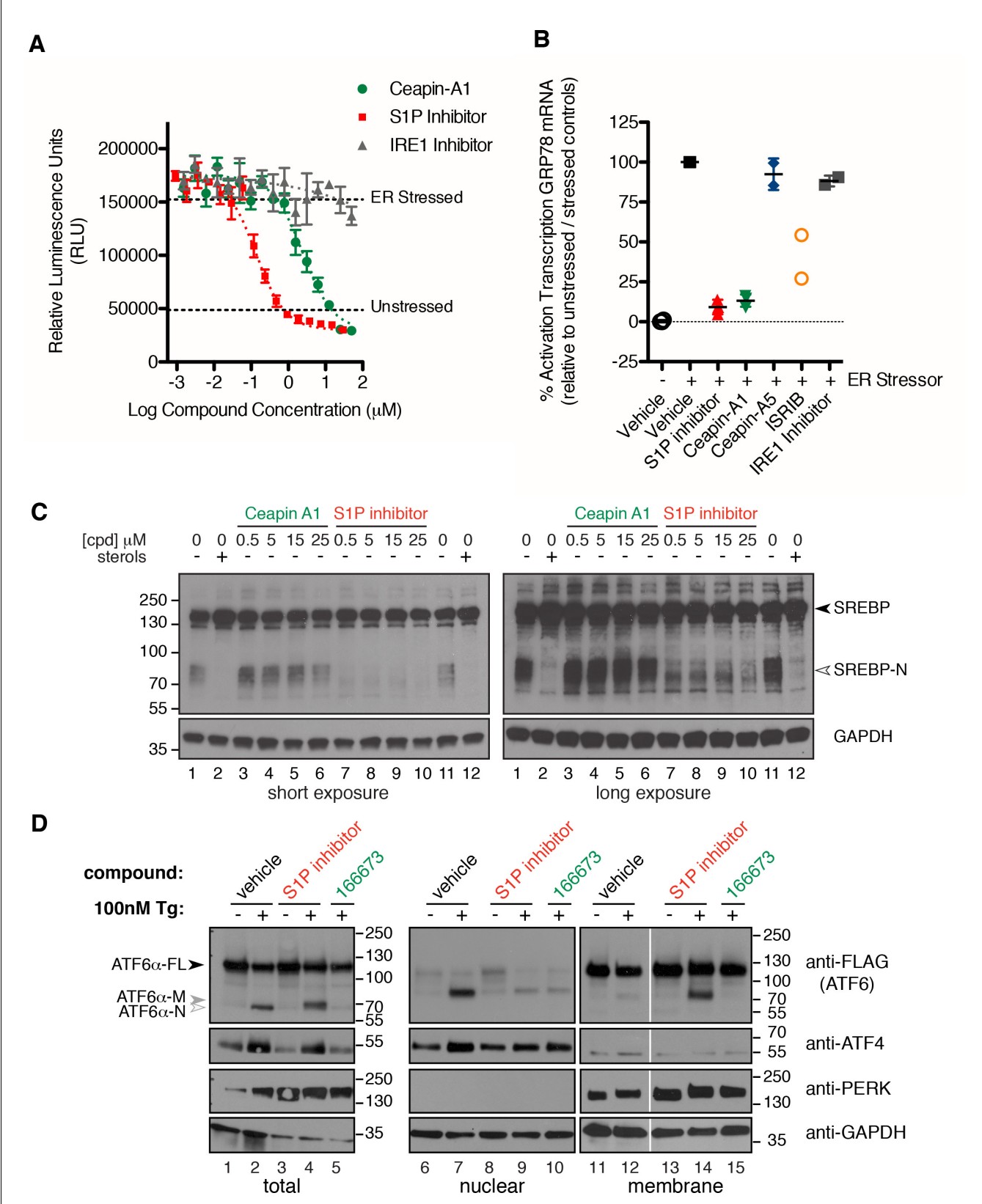

**Figure 3.** Isolation of Ceapin-A1, a small molecule inhibitor of ATF6 but not SREBP processing. (**A**) ERSE-luciferase assay in HEK293T cells. Cells were treated without (DMSO) or with ER stressor (100 nM Tg) in the presence or absence of inhibitors for nine hours. Increasing concentrations of either S1P

*Figure 3 continued on next page*

Figure 3 continued

inhibitor (Pf-429242, red) or Ceapin-A1 (green) but not IRE1 inhibitor (4 µ8C, grey) block ER stress-induced luciferase activity. Plotted is one representative experiment showing mean and standard deviation for each inhibitor concentration (triplicate wells per point). Dashed grey lines indicate the relative luciferase activity of unstressed and stressed controls. (B) ER stress induced upregulation of the endogenous ATF6α target gene GRP78 in U2-OS cells. Cells were treated without (DMSO, open circles) or with ER stress (100 nM Tg, black squares) in the absence or presence of inhibitors for four hours prior to isolation of mRNA. Upregulation of GRP78 mRNA was measured using qPCR. mRNA levels for GRP78 were normalized to GAPDH for each well and then compared to unstressed and stressed controls. ER stress induced GRP78 mRNA induction is inhibited by co-incubation with either S1P inhibitor (2.3 µM Pf-429424, red) or Ceapin-A1 (10 µM, green) but not the inactive Ceapin analog A5 (10 µM, blue). Inhibition of the ISR (200 nM or 400 nM ISRIB, orange) partially inhibits GRP78 induction while inhibition of IRE1 (10 µM 4 µ8C, grey) has only minor effects. Data plotted is the mean percent activation of GRP78 transcription relative to unstressed (0%) and stressed (100%) controls from two or three independent experiments, each with duplicate reactions carried out on duplicate wells. (C) Induction of SREPB processing by lipoprotein depletion in HeLa cells. HeLa cells were grown in lipoprotein deficient media for 16.5 hr prior to addition of either sterols or inhibitors for five hours. One hour prior to lysis proteasome inhibitor (25 µg/mL ALLN) was added to prevent the degradation of the cleaved SREBP-N fragment. Whole cell lysates were analyzed by Western blotting for SREPB1 and GAPDH. Arrowheads denote positions of full-length (SREBP) and cleaved (SREBP-N) variants of SREBP1. Lipoprotein depletion induces cleavage of SREBP (lanes 1, 11) that is inhibited by addition of sterols (10 µg/mL cholesterol, 1 µg/mL 25-hydroxycholesterol, lanes 2, 12) or increasing concentrations of a S1P inhibitor (Pf-429242, lanes 7–10) but not increasing concentrations of Ceapin-A1 (lanes 3–6). Data shown is representative of three independent experiments. (D) Induction of ATF6α processing by ER stress in T-Rex cells expressing FLAG-tagged ATF6α. Arrowheads denote positions of full-length (ATF6α), cleaved membrane-bound (ATF6α-M) and cleaved nuclear (ATF6α-N) variants of ATF6. Cells were treated without (lanes 1,6,11) or with (lanes 2,7,12) ER stressor (100 nM Tg) alone or in combination with either S1P inhibitor (0.75 µM Pf-429242, lanes 3,4,8,9,13,14) or Ceapin-A1 (14.95 µM, lanes 5,10,15) for two hours prior to harvesting. One hour prior to lysis proteasome inhibitor (MG132, 10 µM) was added. Cells were harvested and separated by centrifugation into total, membrane and nuclear fractions and analyzed by Western blot for ATF6α (anti-FLAG), PERK (membrane control), ATF4 (nuclear control) and GAPDH (loading control). Note that totals were run on 10% gels while membrane and nuclear fractions were run on gradient gels to visualize the migration differences between ATF6α-N and ATF6α-M and between PERK and phosphorylated PERK respectively. Data shown is representative of two independent experiments.

The following figure supplements are available for figure 3:

**Figure supplement 1.** Identification of Ceapin-A1, a small molecule that inhibits ATF6 processing in response to ER stress.

**Figure supplement 2.** Screening workflow Summary of screening workflow that lead to the identification of Ceapin-A1 consisting of primary (yellow), secondary (orange) and tertiary (green) screens.

**Figure supplement 3.** Ceapin-A1 inhibits ER stress induced ERSE-luciferase activity ERSE-luciferase assay in HEK293T cells.

**Figure supplement 4.** Mutation of S1P cleavage site in ATF6α leads to production of ATF6α-M upon ER stress.

the screen is shown in *Figure 3—figure supplement 2*). Of the Class 1 inhibitors, only one consistently blocked ATF6α target gene upregulation: Ceapin-A1 ('A1' standing for Analog 1 the founding compound in the series described below).

Ceapin-A1 inhibited both ERSE-luciferase induction by both Tg (*Figure 3A*) and Tm (*Figure 3—figure supplement 3*) and induction of GRP78 mRNA (*Figure 3B*) with an $IC_{50} = 4.7 \pm 1.1$ µM and to the same extent as the S1P inhibitor. Inhibition of IRE1 (*Cross et al., 2012*) had little effect in the same assays (*Figure 3A and B*). In contrast, inhibition of the integrated stress response (ISR) (*Sidrauski et al., 2013*) showed a decrease in ATF6α target gene induction, as previously reported for PERK knockout cells (*Wu et al., 2007*) (*Figure 3B*). By these criteria, Ceapin-A1 behaved like a selective inhibitor of ATF6α.

Since Ceapin-A1 treatment of stressed cells resulted in the same punctate localization of GFP-ATF6α as the S1P inhibitor in the nuclear translocation assay (*Figure 2C and D*), we next investigated if Ceapin-A1 was in fact an S1P inhibitor. To this end, we monitored the processing of endogenous SREBP1 in HeLa cells (*Figure 3C*). Cells grown in lipoprotein-deficient media activate SREBP processing, indicated by the presence of faster migrating, cleaved SREBP-N (*Figure 3C*, lanes 1 and 11). This cleavage was blocked by the addition of either sterols (*Figure 3C*, lanes 2 and 12) or increasing concentrations of the S1P inhibitor (*Figure 3C*, lanes 7–10) to the cell culture media. Addition of Ceapin-A1 to 25 µM (> five times its $IC_{50}$) had no effect on lipoprotein depletion-mediated SREBP processing (*Figure 3C*, lanes 3–6). Therefore, Ceapin-A1 neither inhibits S1P nor S2P and

emerges as the first small molecule inhibitor of ATF6 that does not inhibit SREBP or other pathways known to use these proteases.

We then analyzed the cleavage of ATF6α directly, using a stable HEK293 based cell line that expressed FLAG-tagged ATF6α from an inducible promoter. Induction of ER stress led to cleavage of full-length ATF6α to produce the faster migrating, active transcription factor ATF6α-N (*Figure 3D*, lanes 1 and 2). Surprisingly, in the presence of the S1P inhibitor, ER stress-induced cleavage of ATF6α was not blocked (*Figure 3D*, lanes 3 and 4). This was unexpected, given that inhibition of S1P prevented both nuclear translocation of GFP- ATF6α (*Figure 2C and F*) and upregulation of ATF6α target genes (*Figure 3A and B*, and *Figure 3—figure supplement 1* and *3*). As a first clue on how to resolve the paradox, we observed that the cleavage product, henceforth referred to as ATF6α-M, produced in the presence of the S1P inhibitor migrated more slowly on SDS polyacrylamide gels than the product ATF6α-N produced by ER stress alone.

We next analyzed the subcellular localization of ATF6α-M using differential centrifugation. In contrast to ER stressed cells, where ATF6α was recovered in the membrane fraction (*Figure 3D*, lane 12) and ATF6α-N is in the nuclear fraction (*Figure 3D*, lane 7), both ATF6α and ATF6α-M were recovered in in the membrane fraction in cells treated with ER stress inducer and the S1P inhibitor (*Figure 3D*, lane 14). We therefore denoted this fragment as ATF6α-M, since it remains membrane-bound and as such is incapable of entering the nucleus and activating ATF6α target genes. The transmembrane protein PERK and the transcription factor ATF4 served as controls for membrane and nuclear fractions respectively (*Figure 3D*). We observed the same fragment in cells expressing a variant of ATF6α, in which the S1P cleavage site was mutated (*Figure 3—figure supplement 4*), indicating that an alternate protease generated ATF6α-M and that ATF6α-M was not a substrate for subsequent membrane-release by S2P cleavage. Importantly, cells treated with both ER stress inducer and Ceapin-A1 did not show any ATF6α cleavage product (*Figure 3D*, lanes 5, 10, 15). This difference strongly suggested that the mechanism of inhibition of ATF6α processing by Ceapin-A1 is not via protease inhibition, but that ATF6α is trapped in a place or state where it cannot be cleaved. For this reason, we chose to name the chemical scaffold of the inhibitor 'Ceapin', after the Irish verb 'ceap' meaning to trap.

## Synthesis of a more potent Ceapin scaffold analog: Ceapin-A7

The original hit Ceapin-A1 is an *N*-benzyl pyrazol-4-yl amide (*Figure 4A*). We used the ERSE-luciferase assay to guide structure-activity relationship (SAR) studies aimed at defining the essential pharmacophore within the Ceapin scaffold. We found that all four rings present in Ceapin-A1 are necessary for activity (*Figure 4B*). Thus, deletion of the furan ring (as in Ceapin-A2) led to a loss of detectable activity in the ERSE-luciferase assay. Among several aryl and heteroaryl rings examined in place of the furan, only a simple phenyl ring as in Ceapin-A3 ($IC_{50}$ 6.9 ± 0.7) afforded activity comparable to Ceapin-A1 ($IC_{50}$ = 4.9 ± 1.2 µM). The benzyl substituent on the pyrazole ring was similarly sensitive to modification. Thus, analogs lacking the ortho and para chloro substituents (Ceapin-A4) or bearing methyl in place of the benzyl group (Ceapin-A5) were inactive (*Figure 4B and D*). Further SAR of ring substitution in the benzyl side chain revealed that substitution at both the ortho and para position with spheroid hydrophobes was optimal for activity (*Figure 4C*). Thus, the 2,4-dibromobenzyl analog Ceapin-A6 was about two-fold more potent than Ceapin-A1, while the corresponding bis-trifluoromethyl congener Ceapin-A7 ($IC_{50}$ = 0.59 ± 0.17 µM) was approximately ten-fold more potent than Ceapin-A1 (*Figure 4C and D*). In contrast, analogs bearing a single trifluoromethyl group at either the para (Ceapin-A8) or ortho position (Ceapin-A9) were notably less potent than Ceapin-A7. Additional studies were performed with Ceapin-A7, representing an improved lead from the Ceapin series and with the inactive Ceapin analog A5 as a negative control.

We next confirmed the potency of Ceapin analogs on endogenous ATF6α target induction using qPCR analysis of GRP78, HERPUD1, and ERO1B, an ER oxidoreductase (*Wu et al., 2007*; *Yamamoto et al., 2007*; *Adachi et al., 2008*) (*Figure 4E*). The mean $IC_{50}$ values for the S1P inhibitor, Ceapin-A1 and Ceapin-A7 calculated from dose-response qPCR assays correlated well with the values obtained using the ERSE-luciferase reporter (*Figure 4F*), validating both the analogs and the use of the ERSE-luciferase assay for SAR studies. As expected, the inactive Ceapin analog A5 does not inhibit induction of these targets (*Figure 3B*). The increased $IC_{50}$ values for HERPUD1 reflect the contribution from the IRE1/XBP1 and PERK/ATF4 branches to transcriptional upregulation of this

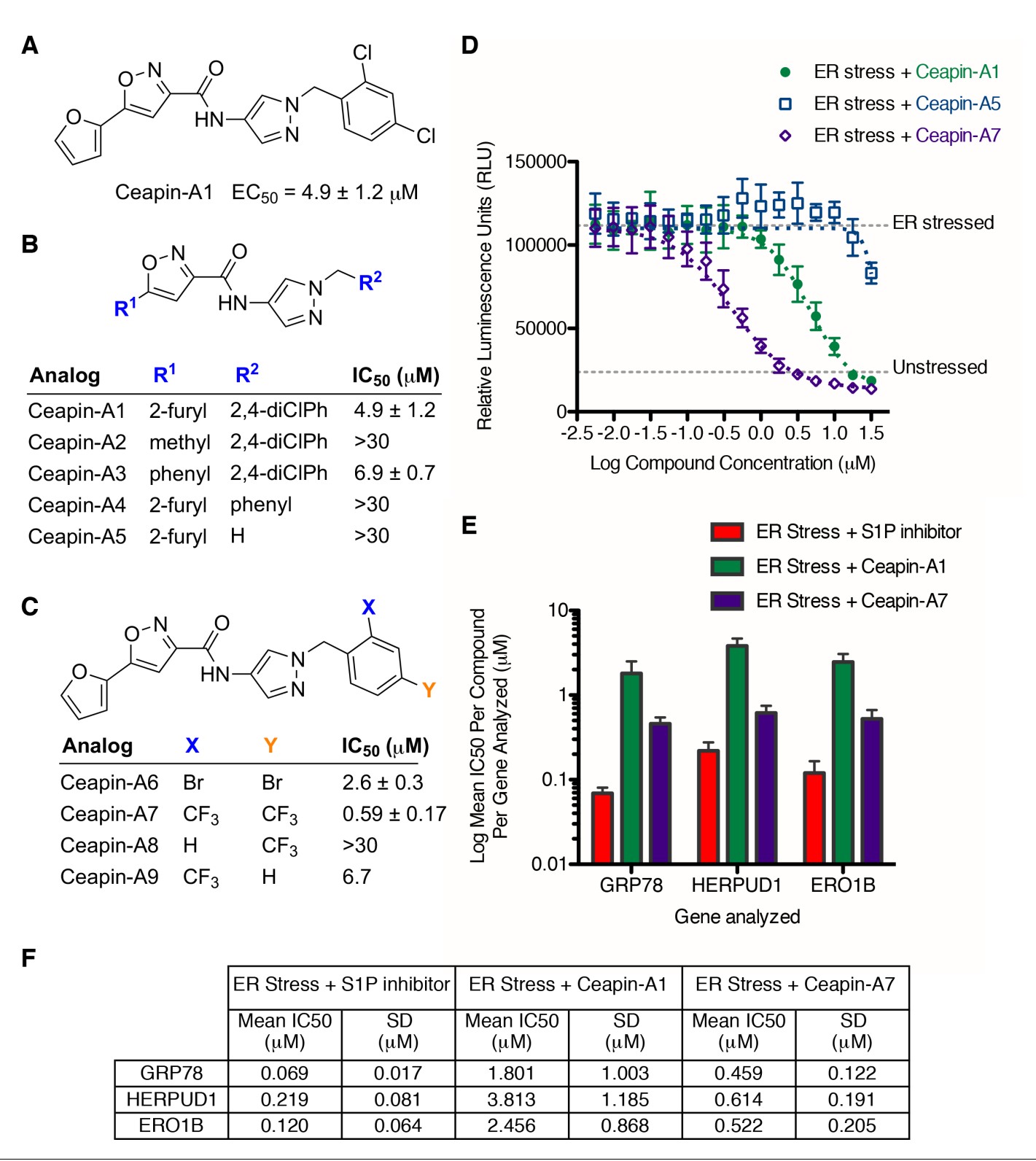

**Figure 4.** SAR studies of Ceapin-A1 improve potency by an order of magnitude. (A–C) Summary of structure activity relationship for Ceapin analogs. (A) Chemical structure of the initial screen hit, Ceapin-A1. SAR of rings (B) or substituents on bis-substituted phenyl ring (C) of Ceapin scaffold. IC50 values were obtained using ERSE-luciferase assay in 293T cells where compounds were tested in dose-response in combination with ER stressor (100 nM Tg). $IC_{50}$ values are from at least four independent experiments for each compound. (D) ERSE-luciferase assay showing improved potency of Ceapin-A7

*Figure 4 continued on next page*

*Figure 4 continued*

(purple) and lack of activity of Ceapin-A5 (blue) compared to Ceapin-A1 (green). 293T cells with stably integrated ERSE-luciferase reporter were treated with ER stressor (100 nM Tg) and increasing concentrations of Ceapin analogs for nine hours prior to reading luciferase activity. Data plotted are mean values from a representative experiment with each point done in triplicate, error bars are standard deviation. (E) qPCR analysis of endogenous ATF6α target genes. U2-OS cells were treated without or with ER stressor in the presence of increasing concentrations of Ceapin analogs for four hours prior to harvesting of mRNA for qPCR analysis. mRNA levels for GRP78, HERPUD1 and ERO1B were normalized to GAPDH for each well and then compared to unstressed controls. Data plotted are mean IC$_{50}$ values calculated from duplicate experiments, each with duplicate qPCR reactions from duplicate wells for S1P inhibitor (Pf-429242, red), Ceapin-A1 (green) and Ceapin-A7 (purple). Error bars are standard deviation. (F) Calculated mean IC50 values and standard deviations from qPCR analysis described above.

target (*Lee et al., 2003*; *Ma and Hendershot, 2004*), further underscoring the selectivity of Ceapin for the ATF6 branch of the UPR.

## Ceapin specifically inhibits ATF6α but not IRE1 or PERK branches of the UPR

Inhibition of IRE1 did not significantly alter ATF6 signaling as seen using ERSE-luciferase or GRP78 mRNA induction upon ER stress (*Figure 3A and B*). We next validated that Ceapins selectively inhibit the ATF6 over the IRE1 and PERK branches of the UPR monitoring the activation of each UPR branch directly. To this end, we used a polyclonal antibody against ATF6α developed by the Mori lab (*Haze et al., 1999*), which allowed us to look at endogenous ATF6α in U2-OS cells (*Figure 5A*). As expected, treatment of cells with Tm produced both a faster migrating unglycosylated form of ATF6α (ATF6α*) and cleaved ATF6α-N (*Figure 5A*, compare lanes 1 and 2). Cells treated with both Tm and Ceapin-A7 contained the unglycosylated form of ATF6α but not ATF6α-N (*Figure 5A*, lane 3), indicating that despite the accumulation of unglycosylated proteins, ATF6α was not cleaved. ATF6α–derived bands from cells treated with Tm and the inactive Ceapin analog A5 were identical to Tm alone (*Figure 5A*, lane 4). ATF6α–derived bands from cells treated with Tg alone or in combination with the inactive Ceapin analog A5 (inactive analog) showed both ATF6α and ATF6α-N (*Figure 5A*, lanes 6 and 8) while cells treated with both Tg and active Ceapin-A7 were indistinguishable from those derived from unstressed cells (*Figure 5A*, lanes 5, 7). Induction of ER stress using either Tg or Tm led to upregulation of BiP protein levels (*Figure 5B*, compare lanes 2 and 6 to lanes 1 and 5). Consistent with our qPCR analysis of its mRNA levels (GRP78 mRNA), Ceapin-A7 but not the inactive Ceapin analog A5 inhibited ER stress-induced upregulation of BiP (*Figure 5B*, compare lanes 3 and 7 to lanes 4 and 8). Thus consistent with our analyses above, Ceapins inhibit cleavage and functional activation of endogenous ATF6α in response to ER stress.

The same lysates were also analyzed for activation of the other branches of the UPR (*Figure 5A*, bottom panels). Ceapin-A7 did not inhibit activation of either the PERK (shown by a slower migrating band representing the phosphorylated form of PERK [*Harding et al., 1999*]) or the IRE1 (shown by production of XBP1s protein) branches of the UPR. These results validate and extend our analyses of the UPR in the 293T-based FLAG-ATF6α reporter cell line. Ceapin-A1 inhibited cleavage of FLAG-ATF6α without inhibiting induction of ATF4 (*Figure 3D*, compare lanes 6, 7 and 10) or autophosphorylation of PERK (indicated by the shift in mobility; *Figure 3D*, compare lanes 11, 12 and 15).

There is cross-talk between UPR branches. Specifically, effectors of both the IRE1 and PERK branches, XBP1 and CHOP respectively, are non-exclusive transcriptional targets of ATF6 (*Wu et al., 2007*; *Adachi et al., 2008*). Treatment of U2-OS cells with ER stressors showed upregulation of both spliced XBP1 mRNA (XBP1s, *Figure 5C*) and DDIT3 mRNA (encodes CHOP, *Figure 5C*). Consistent with Ceapin inhibiting the ATF6-branch selectively, co-incubation of cells with ER stressor and either S1P inhibitor or Ceapin-A1 showed only a partial decrease in upregulation of XBP1s and DDIT3 mRNA (*Figure 5C*, *Figure 5—figure supplement 1*), in agreement with data from ATF6α knockout mouse embryonic fibroblasts (MEF) (*Wu et al., 2007*; *Adachi et al., 2008*). Taken together, these data show that Ceapins are selective inhibitors of the ATF6 branch of the UPR and do not inhibit either IRE1 or PERK signaling.

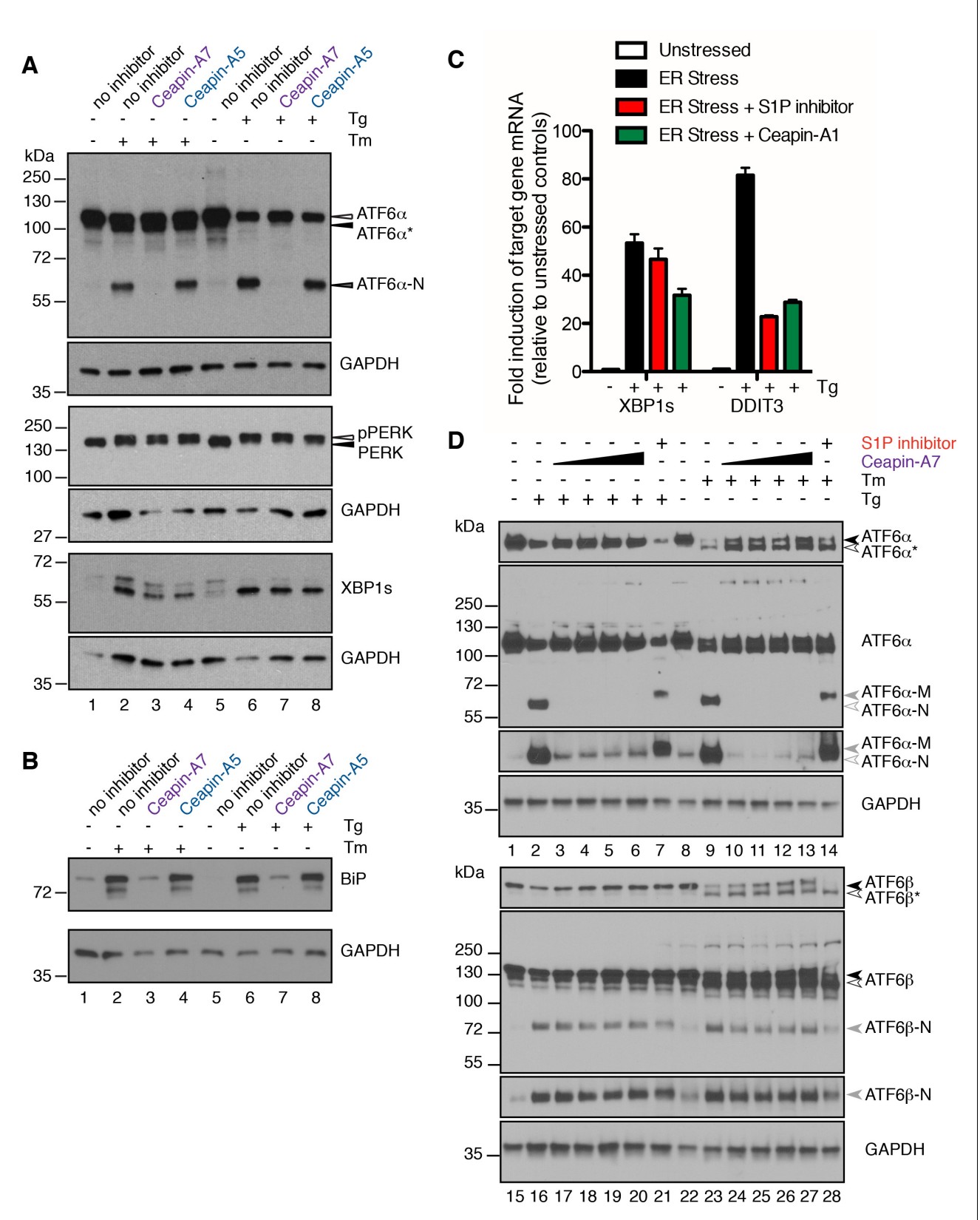

**Figure 5.** Ceapins are selective inhibitors of the ATF6α branch of the UPR. (**A**) U2-OS cells were treated without (DMSO) or with ER stressor (100 nM Tg or 2.5 μg/mL Tm) in the absence or presence of Ceapin analogs (6 μM each) for two hours. One hour prior to lysis, proteasomal inhibitor (10 μM

*Figure 5 continued on next page*

*Figure 5 continued*

MG132) was added to prevent the degradation of the cleaved ATF6α-N fragment. Cells were harvested and analyzed by Western Blot for ATF6α, PERK, XBP1 and GAPDH (loading control). Arrowheads denote the positions of full-length (ATF6α), unglycosylated full-length (ATF6α*) and cleaved (ATF6α-N) variants of ATF6α and also PERK and phospho-PERK. Data shown is representative of three independent experiments. (B) U2-OS cells were treated without (DMSO) or with ER stressor (100 nM Tg or 2.5 µg/mL Tm) in the absence or presence of Ceapin analogs (6 µM each). After eight hours, cells were harvested and analyzed by Western Blot for BiP and GAPDH (loading control). (C) U2-OS cells were treated without (DMSO) or with ER stressor (100 nM Tg, black) in the absence or presence of ten-fold the $IC_{50}$ of either S1P inhibitor (Pf-429242, 3.2 µM, red) or Ceapin-A1 (35.7 µM, green). Four hours later cells were harvested and mRNA extracted. mRNA levels for XBP1s or DDIT3 were normalized to GAPDH for each well and then compared to unstressed controls. Data plotted are from duplicate qPCR reactions from duplicate wells, error bars are standard deviation. (D) U2-OS cells were treated without (DMSO) or with ER stressor (100 nM Tg or 2.5 µg/mL Tm) in the absence or presence of increasing concentration of Ceapin-A7 (0.6, 1.89, 6, 18.9 µM) or S1P inhibitor (5 µM Pf-429242) for four and a half hours. One hour prior to lysis, proteasomal inhibitor (10 µM MG132) was added to prevent the degradation of the cleaved ATF6α-N and ATF6β-N fragments. Cells were harvested and analyzed by Western Blot for ATF6α, ATF6β and GAPDH (loading control). Arrowheads denote the positions of full-length (ATF6α, ATF6β), unglycosylated full-length (ATF6α*, ATF6β*), cleaved membrane bound (ATF6α-M) and cleaved (ATF6α-N, ATF6β-N) variants of ATF6α and ATF6β. Data shown is representative of three independent experiments.

The following figure supplements are available for figure 5:

**Figure supplement 1.** Induction of XBP1s and DDIT3 mRNA is only partially inhibited by either the S1P inhibitor or Ceapin-A1 U2-OS cells were treated without (DMSO) or with ER stressor (2.0 µg/mL Tm, black) in the absence or presence of ten-fold the $IC_{50}$ of either S1P inhibitor (Pf-429242, 3.2 µM, red) or Ceapin-A1 (35.7 µM, green).

**Figure supplement 2.** ATF6β-N is generated in ER stressed cells treated with Ceapin-A7 but not ER stressed cells treated with the S1P inhibitor.

## Ceapin selectively inhibits ATF6α but not ATF6β

The ER contains two related bZip proteins – ATF6α and ATF6β (*Haze et al., 2001*). ATF6α and ATF6β show 41% identity in their amino acid sequences, with both S1P and S2P cleavage sites conserved. ATF6β is activated with similar kinetics and in response to the same stressors as ATF6α and, as ATF6α, moves from the ER to the Golgi apparatus where it is processed by S1P and S2P to release ATF6β-N from the membrane allowing its nuclear translocation. Given the similarity between these related proteins, we next tested if Ceapins inhibit processing of ATF6β in response to ER stress. To this end, we treated U2-OS cells either with Tg or Tm in the absence or presence of increasing concentrations of Ceapin-A7 or with S1P inhibitor and analyzed the migration pattern of both endogenous ATF6α and ATF6β-derived bands (*Figure 5D*). Consistent with our previous results, we observed no ATF6α-N in lysates from Ceapin-A7 treated cells (*Figure 5D*, lanes 3–6 (Tg) and lanes 10–13 (Tm)) but observed ATF6α-M in lysates from S1P inhibitor-treated cells (*Figure 5D*, lane 7 (Tg) and lane 14 (Tm)). In contrast, Ceapin-A7 had no effect on the production of ATF6β-N, even at the highest concentration (18.9 µM; corresponding to 33x its $IC_{50}$ for ATF6α) (*Figure 5D*, lanes 17–20 and 24–27). We further confirmed that while ATF6β-N is not present in nuclear extracts of ER stressed cells treated with S1P inhibitor, cells subjected to ER stress in the absence or presence of Ceapin-A7 both have nuclear localized ATF6β-N (*Figure 5—figure supplement 2*).

## Ceapin sensitizes cells to ER stress

ATF6α knockout mice and MEF cells show impaired survival in the face of acute (*Yamamoto et al., 2007*) or chronic (*Wu et al., 2007*) ER stress. We tested if Ceapins would similarly impair survival of human cells treated with an ER stressor. To this end, we treated U2-OS cells with increasing concentrations of Tg and monitored cell viability over a seventy-two hour time course (*Figure 6A*). From the survival curve, we measured an $IC_{50}$ of 7.1 nM for Tg alone in U2-OS cells (*Figure 6B*). Cells co-incubated with both Tg and Ceapin-A7 showed an almost two-fold increase in sensitivity to ER stress ($IC_{50}$ = 4.5 nM), whereas the inactive Ceapin analog A5 showed no difference (*Figure 6B*). This two-fold difference is consistent with the data from genetic ablation of ATF6α in mice (*Yamamoto et al., 2007*).

Since changes in cell viability may be due to cytostatic and/or cytotoxic effects, we next determined whether Ceapin-A7 displayed increased apoptosis in response to ER stress. To this end, we

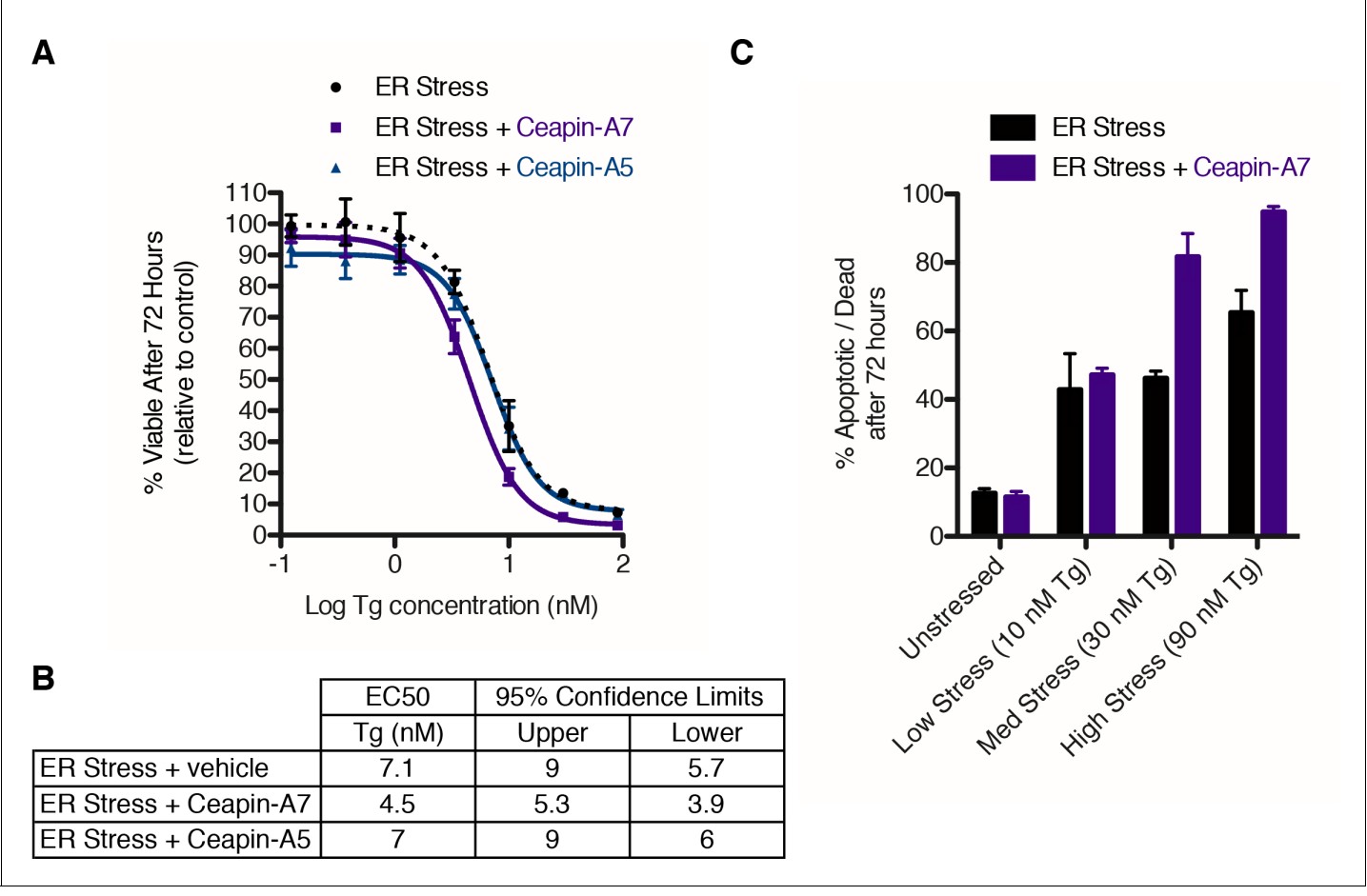

**Figure 6.** Ceapin-A7 sensitizes cells to ER stress. (**A–B**) U2-OS cells were treated with increasing concentrations of ER stressor (Tg) in the absence (black) or presence of six micromolar Ceapin analogs - Ceapin-A5 (inactive, blue) or Ceapin-A7 (purple). (**A**) After seventy-two hours reducing potential of living cells was assayed to determine cell viability. Data plotted are the means of four independent experiments performed in triplicate, error bars represent the standard error of the mean. (**B**) EC$_{50}$ values calculated for ER stressor in the absence or presence of Ceapin analogs showing mean and 95% confidence limits. (**C**). U2-OS cells were treated with increasing concentrations of ER stressor (Tg) in the absence (black) or presence of 6 µM Ceapin-A7 (purple). To analyze cell death, cells were stained with Annexin V and 7AAD and analyzed by flow cytometry. Data plotted are the means from three independent experiments performed in duplicate; error bars represent standard deviation.

stained cells with annexin V, which measures phosphoserine flipping as a marker for apoptotic cells, and 7-aminoactinomycin D (7AAD), a membrane impermeable dye taken up only by cells with compromised plasma membranes as a marker for late apoptotic / necrotic cell death. We treated cells with or without ER stressor at different concentrations in the absence or presence of Ceapin-A7 and analyzed the cells by flow cytometry (*Figure 6C*). Cells treated with Ceapin-A7 alone showed no difference in cell death compared to vehicle alone, consistent with previous work demonstrating that homozygous ATF6α knockout mice are viable and fertile (*Wu et al., 2007*; *Yamamoto et al., 2007*). At low concentrations of ER stress (10 nM Tg), inhibition of ATF6α did not enhance cytotoxicity, however as the concentration of ER stressor was increased (30 nM and 90 nM), ATF6α inhibition resulted in a two-fold increase in apoptotic cells compared to cells treated with ER stressor alone. Thus human cells treated with ER stress and Ceapin-A7 phenocopy the results obtained using genetic ablation of ATF6α in mouse models. Ceapins therefore define a first-in-class series of ATF6α inhibitors that selectively blocks ATF6α and not ATF6β, SREBP or other UPR branches without relying on inhibition of the proteases that are also used by other critical signaling pathways.

## Discussion

In this work we describe the isolation, specificity and chemical refinement of the Ceapin scaffold, the first selective and highly potent pharmacological inhibitors of the ATF6α branch of the UPR. The ATF6α pathway is the least well-understood UPR branch – its mechanism of activation in response to ER stress remains unknown, and the lack of unique enzymes in the pathway precluded development of biochemical assays for screening purposes. Prior to this work, the few compounds shown to inhibit ATF6α signaling (PDI and S1P inhibitors) target important housekeeping enzymes that are shared among multiple pathways and hence elicit pleiotropic effects (*Maurel et al., 2015*). Indeed, ATF6 was considered 'undruggable' (*Maly and Papa, 2014*). Our study highlights the value of combining chemical biology with cell-based screening to identify precise tools for 'intractable' pathways, as amply demonstrated in other pioneering work (*Cassidy-Stone et al., 2008*; *Schreiber et al., 2015*).

To identify Ceapins, we combined cell-based screens to isolate a highly selective inhibitor of ATF6α. Using a broad range of secondary assays for different steps of pathway activation, we built a toolbox of small molecules with which to interrogate ATF6α signaling at different points. We successfully isolated compounds that act both upstream of proteolytic cleavage, corresponding to the early stages of activation of ATF6α, and downstream of nuclear import, corresponding to transcriptional activation. From over 100,000 compounds screened, we found one, Ceapin-A1, that is not only selective for ATF6α but acts precisely at its initial activation stage that currently is the least understood step. As we detail in the accompanying manuscript, Ceapins induce rapid, reversible clustering of ATF6α, preventing exit of ATF6α from the ER (*Gallagher and Walter, 2016*).

Ceapins inhibit activation of ATF6α in response to ER stress without inhibiting ATF6β or SREBP activation, two other ER-bound transcription factors that are similarly trafficked and processed by S1P and S2P in the Golgi apparatus. Ceapins also do not inhibit the IRE1 or PERK branches of the UPR, indicating that the compounds do not generally interfere with sensing ER stress. Further underscoring their high selectivity, Ceapins show no toxicity to unstressed cells, consistent with the fact that ATF6α knockout mice are viable and fertile. Loss of ATF6α becomes detrimental when cells or animals are treated with ER stressors, and Ceapin action mimics this effect. Thus Ceapins promise to be invaluable, first-in-class tools to investigate the role of ATF6α independently or in combination with inhibition of other UPR branches in models of human disease.

Efforts to study the role of ATF6α in disease models have been limited to knockdown or overexpression experiments. ATF6α was shown to be required for the survival of dormant tumor cells (*Schewe and Aguirre-Ghiso, 2008*). In a cystic fibrosis model, knockdown of ATF6α was shown to increase delivery of the poorly folded △F508 variant of the cystic fibrosis transmembrane conductance regulator to the plasma membrane where it could function (*Kerbiriou et al., 2007*). Thus ATF6α inhibitors could be beneficial in diseases where decreased ER quality control would ameliorate cell function. In contrast, enhancing ATF6α activity using inducible activation of ATF6α-N increased quality control of protein folding in the ER and decreased secretion and extracellular aggregation of amyloidogenic proteins involved in light chain amyloidosis (*Cooley et al., 2014*). Ceapins offer a new strategy to investigate the role of ATF6α in existing disease models without requiring the introduction of knockdown or overexpression constructs. While the pharmacokinetic properties of current Ceapin analogs limit their usefulness in animal studies, continued SAR studies to improve metabolic stability offer promise for the future.

Much is made in reviews about the therapeutic potential of modulating the UPR and proteostasis networks (*Ryno et al., 2013*; *Lindquist and Kelly, 2011*; *Brandvold and Morimoto, 2015*). While there has been an explosion in small molecule modulators of the UPR, it is still unclear which kinds of modulations open therapeutic windows for individual disease states. Some cell types exclusively activate one branch, others all three, and often it is not even clear whether down- or up-regulation of one branch or the other would lead to the desired outcome. In the UPR network, extensive crosstalk between the three branches further complicates the issue, with both compensation and interdependence between the UPR branches having implications for how to manage UPR modulation for therapeutic benefit (*Wu et al., 2007*). For example, both PERK and ATF6 act downstream of vascular endothelial growth factor (VEGF) mediated cell survival and angiogenesis (*Karali et al., 2014*). Additionally, inhibition of IRE1 alone in models of diabetes led to hyper-activation of ATF6 that led to severe nephropathy (*Madhusudhan et al., 2015*). As these examples illustrate, appreciating the

level of cross talk and compensation between UPR branches is essential to the design of successful UPR-based therapeutic strategies. To date, this was only possible for IRE1 and PERK. With the identification of Ceapins, we finally have tools to modulate all three branches and to unleash the full potential of UPR modulation that has remained theoretical up to this point.

# Materials and methods

## Plasmid construction

### ERSE-luciferase reporter

A DNA sequence containing 5 copies of the ERSE element: CCAATCGGCGGCCTCCACG (red = NF-Y binding, blue = ATF6 binding) spaced by nine nucleotides was synthesized, PCR amplified with 5' BglII and 3' Acc65I overhangs and cloned using BglII / Acc65I into pGL4.28 (Promega, E846A) which contains a minimal CMV promoter upstream of the luc2CP gene, a synthetically derived luciferase sequence with humanized codon optimization and hCL1 and PEST destabilization sequences. After sequence verification, clones containing two (D9 (=pCGG008), D10), three (D5) or four (D1, D7) copies of the ERSE element were recovered.

These ERSE promoter variants driving luciferase were excised from pGL4.28 by digesting with FseI (to exclude the SV40 polyA terminator), blunting with T4 DNA polymerase, purifying and subsequent digestion with BglII. They were ligated into the retroviral vector pQCXIP (Clontech, 631516) that had been digested with XbaI, blunted with T4 DNA polymerase, purified, digested with BglII and then dephosphorylated. Plasmids were verified by sequencing and two were selected for generation of stable cell lines – 2xERSE-Luciferase (D9 clone 3,) and 3xERSE-luciferase (D5 clone 5).

### MPZ-GFP

The coding region for myelin protein zero (MPZ) was amplified from a pINCY plasmid containing MPZ (Open Biosystems # IHS1380-97434176, LIFESEQ 3361858 NM_000530 - incyte full length human cDNA clone) using oligonucleotides containing 5' HindIII and 3' BamHI sites. Purified PCR product was digested and ligated into HindIII / BamHI linearized pEGFP-N3 (Clontech). The resulting MPZ-monomeric-EGFP fusion was subcloned using HindIII / NotI into HindIII / PspOMI digested dephosphorylated pDEST-FRT-TO (kind gift from Andrew N. Krutchinsky).

### 6xHis-3xFLAG-HsATF6α - wild-type and R416A alleles

The coding region for 3xFLAG-HsATF6α was obtained from pCMV7-3xFLAG-HsATF6a (kind gift from Ron Prywes) (*Shen and Prywes, 2004*). The R416A mutation was introduced by site-directed mutagenesis using a single oligonucleotide 5' - gtgagccctgcaaatcaaaggGCgcaccttctaggattttctgc – 3'. Wild-type or R416A alleles were amplified by PCR using a 5' oligonucleotide containing 6xHIS and attB1 site and 3' oligo with attB1 site and recombined using Gateway technology firstly into the entry vector pDONR-221 using BP clonase (Life Technologies # 11789020) and from there into the destination vector pDEST-FRT-TO using LR clonase (Life Technologies # 11791020).

## Cell line construction and culture conditions

Growth media was DMEM with high glucose (Sigma D5796) supplemented with 10% FBS (Life technologies # 10082147), 2 mM L-glutamine (Sigma G2150), 100 U penicillin 100 µg/mL streptomycin (Sigma P0781). Additional cell line specific supplements are detailed below. Cells were incubated at 37°C, 5% CO2 unless stated otherwise.

Human bone osteosarcoma (U2-OS) cells (ATCC HTB-96) and human embryonic kidney (HEK) 293T cells (ATCC CRL-3216) were obtained from the American Type Culture Collection. U2-OS cells stably expressing GFP-ATF6α were purchased from Thermo Scientific (084_01). Growth media was supplemented with 500 µg/mL G418 (Roche 04 727 878 001) to maintain expression of GFP-ATF6. HeLa-NF cells were a generous gift from Paul Wade (NIH) (*Fujita et al., 2003*). The XBP1 reporter cell line (HEK293T XBP1-Luciferase) was derived from the HEK 293T cell line (ATCC CRL-3216) and was described previously (*Mendez et al., 2015*). The ERSE-luciferase reporter cell line was also derived from the HEK 293T cell line (ATCC CRL-3216) and is described below. 293 T-REx cells expressing doxycycline-inducible 6xHis-3xFLAG-HsATF6α (wild type (*Sidrauski et al., 2013*) or mutant) or MPZ-GFP are derived from (Tet)-ON 293 human embryonic kidney (HEK) cells (Clontech)

containing a ferritin-like protein (Flp) recombination target (FRT) site (*Cohen and Panning, 2007*) and are described below. Commercially available cell lines were authenticated by DNA fingerprint STR analysis by the suppliers. All cell lines were visually inspected using DAPI DNA staining and tested negative for mycoplasma contamination.

### ERSE-luciferase reporter cell line (293T-D9)
Retroviral ERSE-luciferase vectors were used to produce recombinant retroviruses using standard methods. Briefly, pQCXIP-ERSE-Luciferase vectors were co-transfected with a VSV-G envelope on a separate plasmid (Clontech Retro-X Universal Packaging System, 631512) using lipofectamine and optiMem into the GP2-293 packaging cell line grown in antibiotic free, high glucose (4.5 g/L) DMEM supplemented with 1 mM sodium pyruvate, 10% fetal bovine serum and 4 mM L-glutamine. The resulting viral supernatant was harvested at 24 hr and 48 hr and used to transduce HEK293T (ATCC CRL-3216) cells that were then selected with puromycin. The stable cell line generated from the 2xERSE-luciferase construct (D9, PWM112) showed the best fold induction in response to ER stress and was used for the screen and all ERSE-luciferase assays in this manuscript. An early passage of 293T-D9 was expanded and frozen in aliquots such that the same passage of cells was used for each screening day.

### MPZ-GFP and 6xHis-3xFLAG-HsATF6-alpha-R416A T-Rex cell lines
pcDNA5-FRT-TO plasmids were co-transfected with pOG44 into (Tet)-ON 293 human embryonic kidney (HEK) cells containing a ferritin like protein (Flp) recombination target (FRT) site (Flp-In T-Rex cells) (*Cohen and Panning, 2007*) according to manufacturers instructions (Invitrogen). After selection with 100 µg/mL Hygromycin B (Gold Biotechnology) single colonies were isolated, expanded and tested for expression of either tagged MPZ or tagged ATF6-alpha. Expression of MPZ-GFP or 6xHis-3xFLAG-HsATF6-alpha variants was induced with 50 nM doxycycline.

## High throughput small molecule screening
### 384 well ERSE-luciferase assays
106,281 compounds from the UCSF Small Molecule Discovery Center diversity collection (Chem-Bridge Diverse, ChemDiv Diversity and SPECS) were stored as 10 mM stocks in 384w plates at -80°C. 293T-D9 cells were thawed 3–4 days and split once prior to assay. For the primary screen, 10,000 cells in 50 µL were plated per well in poly-D-lysine coated white 384 well plates (Greiner bio-one CELLCOAT 781945) using either Biomek FXp liquid handler (Beckman Coulter) with V&P Scientific Tumble Stir reservoir or WellMate bulk dispenser. Plates were sealed with breathable seals (E&K Scientific AeraSeal T896100-S) and incubated in humidified chambers (Corning 431301 Square Dish with wet tissue lining) for 18–24 hr. 50 nL compound per well was added from DMSO stocks in compound plates (Greiner 784201 E1312150 384w plate PP conical V bottom) using Biomek FXp with 384 pin tool for 10 µM final compound concentration in 0.1% DMSO final. 2 µL ER stress inducer was added and wells were mixed using 384w tips (Fluotics #P30-384FX.NS) for 100 nM thapsigargin final. Plates were sealed with breathable seals and incubated for 9 hr in humidified chambers. Plates were cooled to room temperature for 10 min prior to addition of 13.4 µL One-Glo reagent (Promega #E6130) per well using BioTek EL406 bulk dispenser. Plates were shaken to improve cell lysis and after 5 min incubation were read by Analyst HT plate reader (Molecular Devices). For comparison between days each plate was normalized using unstressed (DMSO, 100% inhibition) and ER stressed (Tg, 0% inhibition) controls (32 wells of each per plate) and the percent inhibition of ER stress induced luciferase for each compound was calculated. Controls were used to calculate fold induction (S/B) and z' score for each plate as follows:

$$\text{S/B} = (\text{mean ER stressed controls})/(\text{mean unstressed controls}).$$
$$\text{Z}' = 1 - (3*(\text{standard deviation stressed ctrls} + \text{standard deviation unstressed ctrls})/$$
$$(\text{mean stressed controls} - \text{mean unstressed controls}))$$

For dose-response assays, compounds were plated in 16 point two-fold serial dilution in DMSO starting at 50 mM in 384w plates and the assay was run as described for the primary screen. Average S/B and z' for secondary screen DR assays were 2.87 ± 0.19 and 0.56 ± 0.09 respectively. Average S/B and z' for tertiary screen DR assays were 2.75 ± 0.13 and 0.64 ± 0.05 (100 nM Tg) and 1.59 ± 0.04

and 0.25 ± 0.16(2 µg / mL Tm) respectively. For structure-activity relationship (SAR) studies cell number was reduced to 6250 cells / well which improved both S/B and z' for the assay.

## 96 well ERSE-luciferase assays

For SAR studies a subset of assays were run in 96w format. The assay was run essentially as described for 384w plates with the following adjustments. Cell plating and compound or one-glo addition was done manually. 20,000 cells in 100 µL final were plated per well (poly-D-lysine coated 96w, Greiner bio-one CELLCOAT 655945). To keep final DMSO concentration at 0.21% compounds were first plated as 500x stocks in DMSO in 16 point quarter log dilution series in microplates (compounds, Applied Biosystems MicroAmp Optical 96w N8010560), diluted to 6x in media containing either ER stress inducer or vehicle in plates (Fisherbrand 96 w 14-230-232) immediately prior to addition to cells for 1x final (20 µL per well). After 9-hr incubation, 31.2 µL of one-glo was added per well and plates were read on a SpectraMax M5 plate reader (Molecular Devices) using Softmax Pro v.5.4.1. The IRE1 inhibitor (4 µ8C) was purchased from Matrix Chemicals.

## 96 well high content assays (nuclear translocation assay and inducible GFP assay)

Image based secondary screens were run in 96 well format. Cells were plated 18–24 hr before induction of ER stress, covered with breathable seals (E&K Scientific AeraSeal T896100-S) and incubated in a humidified chamber at 37°C, 5% $CO_2$. Compounds were added using a quad-mapping protocol from 384w compound plates into 96 well assay plates using Biomek FXp with 96w pin tool. Inducers were added and wells were mixed using 96w Biomek tips (250 µL tip for Biomek FX, USA Scientific # 1062–2410). Plates were imaged in an IN Cell Analyzer 2000 (GE Healthcare Life Sciences) using 20x air objective and Cy3, DAPI and FITC excitation and emission filtersets. Images were analyzed using CellProfiler and MATLAB (see below).

## Inducible GFP assay (MPZ-GFP in 293 T-Rex cells) – compound treatment and imaging

For MPZ-GFP T-Rex cell-line, 96 well plates (Greiner Costar 3595) were coated overnight with 100 µL of 10 µg/mL fibronectin in PBS (Sigma F1141) at 37°C. 150 µL of $0.1 \times 10^6$ cells/mL were plated per well. For inducible GFP assay compounds were tested at three doses – 6.6, 2.2 and 0.7 µM final – in the presence of 50 nM doxycycline (Sigma D9891) and incubated for eight hours prior to fixation. Average z' for duplicate experiments was 0.74 ± 0.12. Inhibitors used were cycloheximide (Sigma C7698) and S1P inhibitor (Pfizer Pf-429242).

Media was aspirated and 100 µL of fixative (4% paraformaldehyde in PBS, P6148, Sigma) was added per well and incubated 10 min at room temperature (RT). After fixation, cells were washed three times with PBS, permeabilized with PBS containing 0.1% Triton-X100, washed with PBS and then blocked with PBS containing 2% normal goat serum (Jackson ImmunoResearch Laboratories # 005-000-121) for one hour at RT. Primary antibody (anti-GFP, mAb 3E6 Invitrogen # A11120) was used at 1:1000 dilution in blocking solution and incubated at 4°C overnight. Cells were washed three times with PBS then incubated with secondary antibody (anti-mouse-Alexa-488, Life Technologies # A11029) and rhodamine phalloidin in blocking solution for two hours RT. Cells were washed three times with PBS, the first wash containing Hoechst 33,342 (Invitrogen # H1399) at 1:1000 dilution. Washing and permeabilization steps were five minutes each.

## Inducible GFP assay (MPZ-GFP in 293 T-Rex cells) – image analysis

Image analysis used CellProfiler. After importing of images, illumination correction for each channel was performed using background algorithm with block size of 120 without smoothing. Nuclei were identified from the Hoechst image as primary objects with a diameter of 15–40 pixel units. Cells were identified from the actin image as secondary objects using propagation from the primary objects using global Otsu three-class thresholding with weighted variance – middle intensity pixels were assigned to the foreground. GFP intensity for each cell was calculated from the original MPZ-GFP image and exported as a MATLAB file for analysis.

## Nuclear translocation assay (GFP-ATF6α in U2-OS cells) – compound treatment and imaging

GFP-ATF6α U2-OS cells were plated in media containing G418 selection agent in 96 well plates – 100 µL of $0.032 \times 10^6$ cells per mL in ibidi 96-well ibiTreat µ-plate (ibidi 89626). Compounds were tested at 6.6 µM final in the presence of ER stress inducer (100 nM Tg) and incubated for five hours prior to fixation. Average z' for triplicate experiments was $0.70 \pm 0.18$.

After 5 hr, media was removed and cells were fixed in 4% PFA (Electron Microscopy Sciences 15714) in PHEM buffer (60 mM PIPES, 25 mM HEPES, 10 mM EGTA, 2 mM $MgCl_2$, pH 6.9) for 15 min RT. Cells were permeabilized with PHEM-Tx (PHEM containing 0.1% Triton X-100, two washes, 5 min RT), washed twice in PHEM, blocked in PHEM containing 2% normal goat serum (Jackson Immunoresearch Laboratories 005-000-121) for 1 hr at RT. Primary antibodies were incubated in blocking solution overnight at 4°C. Cells were washed three times in PHEM-Tx then incubated with secondary antibodies and nuclear stain (DAPI, Molecular Probes D-1306, 5 µg/mL) in blocking solution for 2 hr RT protected from light. Cells were washed three times PHEM-Tx, twice PHEM. Antibodies used were rat anti-GRP94 9G10 (abcam ab2791), mouse anti-GFP 3E6 (Invitrogen A11120), anti-rat-Alexa-555 (Invitrogen A21434), anti-mouse-Alexa-488 (Invitrogen A11029), each at 1:1000 dilution.

## Nuclear translocation assay (GFP-ATF6α in U2-OS cells) – image analysis

Image analysis used CellProfiler. After importing of images, illumination correction for each channel was performed using background algorithm with block size of 120 with smoothing using median filter of 50 pixel unit size. Nuclei were identified from the DAPI image as primary objects with a diameter of 23–43 pixel units. Clumped objects were distinguished based on fluorescence intensity. Resulting objects were shrunk by 1 pixel to prevent the nuclear envelope being counted as nuclear signal. The ER for each cell was identified from the GRP94 image as secondary objects using propagation from the primary objects using global Otsu two-class thresholding with weighted variance. The nuclear area of each cell was subtracted from the ER area to give the final ER mask (see *Figure 2—figure supplement 1*). GFP intensity for the nucleus and ER of each cell was calculated from the original GFP-ATF6α image and the nuclear to ER ratio of GFP intensity was calculated per cell and exported as a MATLAB file for analysis.

## Automated image quantification and hit determination for high-content assays

Data from CellProfiler analysis were exported to MATLAB (R2009a, MathWorks) for further computation.

## Nuclear translocation assay – GFP-ATF6

The values of the ratio of nuclear intensity to endoplasmic reticulum intensity (nuclear-to-ER ratio) for each well and plate were re-ordered to compile all data from one experiment in one single matrix. Both negative (untreated cells (vehicle only)) and positive (thapsigargin treated cells) controls were included in each experiment. Since these conditions are the reference for the rest of the wells in each plate, outliers were removed in order to make the analysis more reliable. Control wells that had a p-value of 0.1 or greater in a two-tailed two-sample t-test (each particular sample tested against the group of controls) were considered as outliers and were removed from further analysis. While most of the experiments showed very homogeneous controls, there was a single plate in which 1 well was disregarded.

To quantify the degree of *activation* (nuclear localization) of each compound, we calculated a nuclear-to-ER ratio threshold above which a cell is classified as activated. The minimum and maximum values in a whole plate were used to define the number of intervals (each one of a range of 0.025 units) used to generate histograms of the distribution of values in the controls. The data from the individual positive (thapsigargin treated) and negative (untreated) control wells were merged into two groups of measurements and the relative frequency of values in each interval for each group was calculated. The intersection of the histograms (relative frequency vs. interval mean value) was defined as the mean value of the interval in which the counting in the positive control was greater than the negative control. This *threshold* was used to classify each cell in a well as *activated*

(with nuclear to cytoplasm ratio greater than the threshold) or *non-activated* (nuclear to cytoplasm ratio less or equal to the threshold). The *percentage of activation* under each treatment was calculated as the percentage of cells in the *activated* state in each well.

The *percentages of activation* in the controls allowed us to define a *percentage of inhibition*. The percentage of activation in the positive control was set to 0% inhibition and the percentage of activation in the negative control was set to 100%. The *percentage of inhibition* (%inh) was calculated for each plate (i) as

$$\%\mathrm{inh_i} = \left(1 - \frac{\%\mathrm{act_i} - \%\mathrm{act_u}}{\%\mathrm{act_t} - \%\mathrm{act_u}}\right) \times 100 \tag{1}$$

where $\%\mathrm{act_i}$ is the *percentage of activation* in well i; $\%\mathrm{act_u}$ and $\%\mathrm{act_t}$ are the mean *percentage of activation* in the untreated controls or the treated controls, respectively.

While the *percentage of inhibition* can be used for scoring, a binary classification was necessary to identify hits in the screening. Conditions in which the inhibition percentage was at least 3 standard deviations greater than the negative control (*cutoff* value) were flagged as *inhibition hits*.

## Inducible GFP assay – MPZ-GFP

The mean fluorescence intensity for each well was imported to Matlab and similar computations were conducted. The mean fluorescence was log-transformed in order to get Gaussian distributions. A two-tailed two-sample t-test was performed in each group of controls (positive and negative) to detect outliers (p-value of 0.05 or grater) and remove them from the pool of control wells.

A percentage of inhibition was calculated by assuming the mean of the logarithm of the mean of the controls to be 0% (untreated cells) or 100% (treated cells) as

$$\%\mathrm{inh_i} = \left(1 - \frac{\log(\mathrm{F})_i - \log(\mathrm{F})_u}{\log(\mathrm{F})_t - \log(\mathrm{F})_u}\right) \times 100 \tag{2}$$

where $\log(\mathrm{F})_i$ is the *logarithm of mean fluorescence* in well i; $\log(\mathrm{F})_u$ and $\log(\mathrm{F})_t$ are the mean *logarithm of mean fluorescence* in the untreated controls or the treated controls, respectively.

In order to identify the compounds that inhibited the expression of MPZ-GFP, we performed a one-tailed two-sample t-test on the logarithm of the fluorescence of each of the wells against the positive (doxycycline induced) control wells (left-tailed and 0.5% confidence). The p-values for each test were used to score compounds by general inhibition. An approximate value for the percentage of inhibition that was detected as significantly different in each well (*cutoff* value) was calculated as the mean of the percentage of inhibition of the minimum hit and the maximum non-hit.

## Determination of toxicity – both nuclear translocation and inducible GFP assays

To detect toxic compounds in both assays, the number of cells in each well was used as a proxy for cell survival. The relative number of cells in each well to the mean of the number of cells in the negative control wells was calculated. A condition was flagged as *toxic* if the relative number of cells was less than 0.5.

## Assessment of the quality of each assay

In order to assess the robustness of each assay plate, we calculated the *z' factor* as

$$z' = 1 - \left(3 \times \frac{(\sigma_t + \sigma_u)}{(\mu_t + \mu_u)}\right) \tag{3}$$

where $\sigma_t$ and $\sigma_u$ are the standard deviations of the treated and untreated controls; $\mu_t$ and $\mu_u$ are the means of the controls (*percentage of activation* in ATF6-inhibition screening and *logarithm of mean fluorescence* in general inhibition screening).

## Quantitative PCR analysis of mRNA expression to measure ATF6α target gene induction

At 18–24 hr prior to drug treatment, 12,000 U2-OS cells were plated per well of a 96 well plate (Costar 3595), covered with breathable seals (Aeraseal) and incubated in a humidified chamber 37°C, 5% $CO_2$. Each compound was tested at in duplicate wells using either 100 nM Tg or 2 µg/mL Tm as ER stress inducer. Eight unstressed and stressed wells (four per inducer) per plate were included as controls. Quarter-log serial dilutions of inhibitors in DMSO at 500x assay concentration were prepared in 96w plates = 'compound plate' (Applied Biosystems MicroAmp Optical 96-well reaction plate N8010560). Media without or with inducer was prepared to 6.073x and added to 96w plates to which 500x inhibitor stocks were added to 6x final = 'inducer plate'. Media without inducer contained DMSO as vehicle for unstressed control. Media containing either vehicle, ER stress inducer or ER stress inducer and inhibitor was added to cells to 1x final, covered with breathable seals and incubated for four hours at 37°C, 5% $CO_2$. The final volume of DMSO was equal between all wells (0.2%). Cells were lysed, RNA was prepared and PCR reactions were assembled using the Power SYBR Green Cells-to-CT kit (Life Technologies #4402955) according to manufacturers instructions. Oligos used for qRT-PCR were as follows:

GRP78 (BiP): 5'-CATGGTTCTCACTAAAATGAAAG-3' and 5'-GCTGGTACAGTAACAACTG-3'.
Herpud1: 5'-CAGAAATCAACGCCAAGGTG-3' and 5'-GAACTTCCCTTTGCCTTAAACC-3'
Ero1LB: 5'-AATCTGAAGCGACCTTGTCC-3' and 5'-GCCCAGCTTTTATTCCAACC-3'
XBP1s: 5'-GGAGTTAAGACAGCGCTTGG-3' and 5'-CCTGCACCTGCTGCG-3'
DDIT3 (CHOP): 5'-AGCCAAAATCAGAGCTGGAA-3' and 5'-TGGATCAGTCTGGAAAAGCA-3'
GAPDH: 5'-TGGAAGATGGTGATGGGATT-3' and 5'-AGCCACATCGCTCAGACAC-3'

For each experiment, duplicate reactions were performed on duplicate wells giving four values for each dose. qRT-PCR reactions were run using a CFX96 Real Time System (Bio-Rad). Expression was normalized first to GAPDH internal control and then compared to stressed controls using CFX Manager 3.0 software (Bio-Rad). For each inhibitor data was log transformed, dose-response curves (log(inhibitor) versus response, variable slope, four parameter) were plotted and IC50 were calculated using Prism 5.Of (GraphPad Software, Inc).

## Analysis of SREBP cleavage in response to lipoprotein depletion

Cleavage of SREBP in HeLa cells was analyzed essentially as described (*Hua et al., 1995*; *Espenshade et al., 1999*; *Sakai et al., 1996*). Briefly, two days prior to drug treatment HeLa cells were plated in growth media at a density of $4 \times 10^4$ cells per well of a six-well plate. The following day, 16.5 hr prior to drug treatment, growth media was replaced with lipoprotein deficient media – DMEM with high glucose (Sigma D5796), 10% lipoprotein deficient serum from fetal calf (LPDS, Sigma S5394), 50 µM compactin (aka mevastatin, a HMG-CoA reductase inhibitor, Santa Cruz Biotechnology sc-200853), 50 µM mevalonolactone to facilitate non-sterol isoprenoids (Sigma 68519) (*Goldstein and Brown, 1990*), 2 mM L-glutamine (Sigma G2150), 100 U penicillin 100 µg /mL streptomycin (Sigma P0781).

After 16.5 hr in lipoprotein deficient media, sterols or inhibitors were added to cells. Sterols added were cholesterol (10 µg/mL, Sigma C3045) and 25-hydroxycholesterol (1 µg/mL, Sigma H1015). Serial dilutions of inhibitors in DMSO at 1000x were prepared for both Ceapin and the S1P inhibitor, Pf-429242. Final concentrations of inhibitors on cells were 0.5, 5, 15 and 25 µM respectively. For each well not receiving either sterols or inhibitors the corresponding vehicle was added such that the final concentration of ethanol and DMSO was equal for all wells. Four hours after addition of inhibitors, a proteasome inhibitor (25 µg/mL ALLN, Sigma A6185) was added to prevent degradation of cleaved SREBP-N. One hour later cells were harvested and protein lysates were prepared.

For lysis, 0.5 mL of scraping buffer was added per well. Cells from each well were scrapped into eppendorf tubes and centrifuged at 3000 g for five minutes at four degrees. Each cell pellet was resuspended in 10 µL of lysis buffer and incubated on ice for twenty minutes. Tubes were then vortexed for five minutes at four degrees to shear genomic DNA, incubated on ice for five minutes, centrifuged at 1000 g for two minutes at four degrees. Total protein concentration per sample was determined from 2 µL of each sample using the BCA assay according to manufacturers instructions (Thermo Scientific 23225). For each sample, 8.92 µg total protein was loaded per lane of a fifteen

well SDS-PAGE gel. After western blotting, membranes were probed with anti-SREBP1 (abcam ab3259) or anti-GAPDH (abcam ab9485).

Scraping buffer is 10 μM MG132 (Sigma C2211), 1x complete protease inhibitor (Roche Diagnostics 05056489001) in phosphate buffered saline (Sigma D8537). Lysis buffer is 200 mM Tris pH 8.0, 1% SDS, 100 mM NaCl, 10 μM MG132, 1x complete protease inhibitor. Loading buffer was added to each sample from a 5x stock to 1x final. 1x loading buffer is 40 mM Tris pH 8.0, 0.2% SDS, 8 mM DTT, 6% glycerol, 10 μM MG132, 1x complete protease inhibitor, bromophenol blue.

## Differential centrifugation of 6xHis-3xFLAG-HsATF6-alpha expressing T-Rex cells

### Cell plating and drug treatment

Two days prior to drug treatment $2.1 \times 10^6$ 6xHis-3xFLAG-HsATF6α T-Rex cells per plate were plated in 100 mm dishes. The following day, expression of tagged ATF6α was induced using 50 nM doxycycline. 22.5 hr later ER stressor (100 nM Tg in DMSO, Sigma T9033) with or without inhibitors was added to cells and incubated for one hour. Vehicle was added to ensure the final concentration of DMSO was the same for all samples. Inhibitors used were S1P inhibitor (0.75 μM Pf-429242, Pfizer) or Ceapin-A1 (5 μM Ceapin-A1).

### Differential centrifugation

To harvest cells, 2.9 mL of scraping buffer was added per plate and cells were scraped into 15 mL Falcon tubes. Samples were centrifuged at 3000 rpm in Beckman GH3.8 rotor at four degrees for fifteen minutes. Cell pellets were resuspended in 1.6 mL Buffer A and incubated for ten minutes on ice. Cells were lysed by passing through a 22.5 gauge needle attached to a 1 mL syringe thirty times. 0.1 mL sample was taken (= total). Samples were spun 1000 g for seven minutes at four degrees. The pellet from this spin is the nuclear fraction and the supernatant contains membranes and cytosol. The nuclear pellet was washed once with Buffer A, resuspended in 0.29 mL buffer B and incubated rotating for one hour at four degrees. Nuclear samples were centrifuged at 100,000 g in Beckman TLA 100.2 rotor for 30 min at four degrees. The supernatant from this spin is the nuclear extract. The membrane and cytosol samples were centrifuged at 100,000 g in Beckman TLA 100.2 rotor for 30 min. The pellet from this step contains membranes, the supernatant is the cytosol. The membrane pellet was washed once in Buffer B and resuspended in 0.29 mL Buffer B. The cytosol containing supernatant was precipitated with five volumes of ice-cold acetone for ten minutes on ice then centrifuged at 3500 rpm in Beckmann GS-6KR for fifteen minutes at four degrees. The pellet from this step was resuspended in 0.29 mL Buffer B. Protein concentration for membrane, nuclear and cytosolic samples was determined using the BCA assay according to manufacturers instructions (Thermo Scientific 23225). For western blot analysis, 15 μL of total, 5 μg membranes or 15 μg nuclear extract was loaded per lane of a fifteen well SDS-PAGE gels (Tg – Any kD, BFA – 10% Bio-rad TGX minigels). After blotting, membranes were probed with the following antibodies: FLAG (1:1000, Sigma M2 F1804), PERK (1:500, Cell Signaling Technology C33E10), ATF4 (1:1000, Santa Cruz sc-200), GAPDH (1:1000, abcam ab-9485).

### Buffers

Scraping buffer: 10 μM MG132 (Sigma C2211), 1x complete protease inhibitor (Roche Diagnostics 05056489001) in phosphate buffered saline (Sigma D8537).

Buffer A: 10 mM HEPES-KOH pH 7.4, 250 mM sucrose, 10 mM KCl, 1.5 mM $MgCl_2$, 1 mM Na-EDTA, 1 mM Na-EGTA, 1x complete protease inhibitor.

Buffer B: 10 mM HEPES-KOH pH 7.6, 2.5% glycerol, 420 mM NaCl, 1.5 mM $MgCl_2$, 1 mM Na-EDTA, 1 mM Na-EGTA, 1x complete protease inhibitor.

SDS loading buffer (5x): 150 mM Tris-HCL pH 7.4, 3% SDS, 5% glycerol, 2.5% β-mercaptoethanol, pinch bromophenol blue.

## Differential centrifugation of U2-OS cells for endogenous ATF6α and ATF6β

### Cell plating and drug treatment

Two days prior to drug treatment $4.13 \times 10^5$ U2-OS cells per plate were plated in 100mm dishes, four dishes per drug treatment. Two days later, ER stressor (100 nM Tg in DMSO, Sigma T9033) with or without inhibitors was added to cells and incubated for four hours. Vehicle was added to ensure the final concentration of DMSO was the same for all samples. Inhibitors used were S1P inhibitor (5 µM Pf-429242, Pfizer) or Ceapin-A7 (6 µM Ceapin-A7). One hour prior to lysis, proteasomal inhibitor (10 µM MG132 (Sigma C2211)) was added to prevent the degradation of the cleaved ATF6α-N and ATF6β-N fragments.

### Differential centrifugation

Differential centrifugation was performed as described above for 6xHis-3xFLAG-ATF6a expressing T-Rex cells except the buffer volumes were quartered. For western blot analysis, 10 µL of total or 15 µg nuclear extract was loaded per lane of a fifteen well SDS-PAGE gels, one gel each for ATF6α and ATF6β. After blotting, membranes were probed with antibodies against ATF6α (*Haze et al., 1999*) (1:1000 in 5% BSA) or ATF6β (*Wu et al., 2007*) (1:1000 in 5% milk) at four degrees overnight. After developing for ATF6α and ATF6β membranes were stripped for sixty seconds shaking at room temperature in stripping buffer (7 M Guanidine hydrochloride (Sigma G4505), 50 mM Glycerol (MP Biomedicals 800689), 50 µM EDTA (Fisher BP120-1), 100 µM potassium chloride (Fisher P217-3), 20 mM β-mercaptoethanol (Sigma M6250), washed twice in distilled water, stripped again for sixty seconds, washed extensively with distilled water, PBS-Tween, blocked in 5% milk before cutting the membranes to probe for PERK (1:500, Cell Signaling Technology C33E10) or ATF4 (1:1000, Santa Cruz sc-200) each in 5% milk in PBS-Tween.

## Synthesis of ceapin analogs

N-{1-[(2,4-Dichlorophenyl)methyl]-1H-pyrazol-4-yl}-5-(furan-2-yl)-1,2-oxazole-3-carboxamide (Ceapin-A1) was purchased from Chemdiv. N-{1-[(2,4-Dichlorophenyl)methyl]-1H-pyrazol-4-yl}-5-methyl-1,2-oxazole-3-carboxamide (Ceapin-A2) andN-{1-[(2,4-dichlorophenyl)methyl]-1H-pyrazol-4-yl}-5-phenyl-1,2-oxazole-3-carboxamide(Ceapin-A3) were purchased from ChemBridge. N-(1-Benzyl-1H-pyrazol-4-yl)-5-(furan-2-yl)-1,2-oxazole-3-carboxamide (Ceapin-A4) and 5-(furan-2-yl)-N-(1-methyl-1H-pyrazol-4-yl)-1,2-oxazole-3-carboxamide(Ceapin-A5) were purchased from Enamine. Reagents and solvents were purchased from Sigma- Aldrich, Acros, Combi-Blocks, AK Scientific, ChemBridge, Enamine or TCI America and used as received unless otherwise indicated. Flash column chromatography was carried out using a Biotage Isolera Four system and Silia*Sep* silica gel cartridges from Silicycle. Hydrogenation reactions were carried out in ThalesNano H-Cube reactor using 30 mm 10% Pt/C catalyst cartridges. $^1$H NMR spectra were recorded on a Varian INOVA-400 400MHz spectrometer. Chemical shifts are reported in δ units (ppm) relative to residual solvent peak. Coupling constants (*J*) are reported in hertz (Hz). Characterization data are reported as follows: chemical shift, multiplicity (s=singlet, d=doublet, t=triplet, q=quartet, br=broad, m=multiplet), coupling constants, number of protons, mass to charge ratio. LC/MS analyses were performed on a Waters Micromass ZQ/Waters 2795 Separation Module/Waters 2996 Photodiode Array Detector/Waters 2424 Evaporative Light Scattering Detector system. Separations were carried out on an XTerra MS $C_{18}$ 5 µm 4.6 × 50 mm column at ambient temperature using a mobile phase of water-methanol containing 0.1% formic acid.

## Method A for alkylation

To a solution of 4-nitro-1H-pyrazole (1 equiv) in N,N-dimethylformamide, were added potassium carbonate (2 equiv) and the benzyl bromide (1 equiv). The mixture was stirred at ambient temperature until judged complete by LC/MS. The reaction mixture was then diluted with ethyl acetate (10 mL), washed with saturated ammonium chloride solution (10 mL), water (10 mL) and brine (10 mL). The organic layer was dried over magnesium sulfate, concentrated under reduced pressure and purified by flash column chromatography (ethyl acetate/hexanes) to obtain the product.

## Method B for hydrogenation

A methanolic solution of the nitro compound was passed through 10% Pt/C catalyst at a rate of 1 mL/min in the H-Cube reactor under atmospheric pressure and ambient temperature until the reaction was judged complete by LC/MS. The reaction mixture was concentrated under reduced pressure to obtain the crude amine that was used without further purification.

## Method C for amide coupling

To a solution of the carboxylic acid (1 equiv) in N,N-dimethylformamide, were added HATU (1.1 equiv), the amine (1 equiv), and N,N-diisopropylethylamine (2 equiv). The mixture was stirred at ambient temperature until the reaction was judged complete by LC/MS. The reaction mixture was then diluted with ethyl acetate (10 mL), washed with saturated ammonium chloride solution (10 mL), water (10 mL) and brine (10 mL). The organic layer was dried over magnesium sulfate, concentrated under reduced pressure and purified by flash column chromatography (ethyl acetate/hexanes) to obtain the product.

## 1-{[2,4-Dibromophenyl]methyl}-4-nitro-1H-pyrazole

**Chemical structure 1.** 1-{[2,4-Dibromophenyl]methyl}-4-nitro-1H-pyrazole.

To a cooled (0°C) solution of 2,4-dibromobenzyl alcohol (0.1 g, 0.37 mmol) in dichloromethane (1.0 mL) and N,N-diisopropylethylamine (0.13 mL, 0.75 mmol) was added dropwise methanesulfonyl chloride (0.032 mL, 0.41 mmol). The mixture was stirred at 0°C for 15 min followed by addition of potassium carbonate (0.1 g, 0.75 mmol) and a solution of 4-nitro-1H-pyrazole (0.042 g, 0.37 mmol) in N,N-dimethylformamide (0.5 mL). The mixture was stirred at 50°C for 18 hr. The reaction mixture was then diluted with ethyl acetate (10 mL), washed with saturated ammonium chloride solution (10 mL), water (10 mL) and brine (10 mL). The organic layer was dried over magnesium sulfate, concentrated under reduced pressure and purified by flash column chromatography (20% ethyl acetate/hexanes) to obtain 0.43 g (82%) of the title compound as a cream colored solid. $^1$H NMR (400 MHz, CDCl$_3$) δ 8.15 (s, 1H), 8.08 (s, 1H), 7.79 (S, 1H), 7.48 (d, $J$ = 8.2 Hz, 1H), 7.12 (d, $J$ = 8.2 Hz, 1H), 5.37 (s, 2H). LCMS $m/z$ 361 (MH+).

## N-(1-{[2,4-Dibromophenyl]methyl}-1H-pyrazol-4-yl)-5-(furan-2-yl)-1,2-oxazole-3-carboxamide (Ceapin-A6)

**Chemical structure 2.** N-(1-{[2,4-Dibromophenyl]methyl}-1H-pyrazol-4-yl)-5-(furan-2-yl)-1,2-oxazole-3-carboxamide (Ceapin-A6).

To a refluxing mixture of 1-{[2,4-dibromophenyl]methyl}-4-nitro-1H-pyrazole (0.06 g, 0.17 mmol) and ammonium chloride (0.09 g, 1.7 mmol) in a 2:1 mixture of ethanol/water (6.0 mL), was added in portions, iron (0.028 g, 0.5 mmol) over a period of 30 min. After refluxing for an additional 5 hr and cooling to ambient temperature, dichloromethane (20 mL) was added to the reaction

mixture. The organic layer was washed with brine, dried over magnesium sulfate and concentrated to obtain 1-{[2,4-dibromophenyl]methyl}-1H-pyrazol-4-amine which was used without further purification.

To a solution of 5-(furan-2-yl)-1,2-oxazole-3-carboxylic acid (0.016 g, 0.09 mmol) in N,N-dimethylformamide (0.5 mL), were added HATU (0.038 g, 0.099 mmol), 1-{[2,4-dibromophenyl]methyl}-1H-pyrazol-4-amine (0.03 g, 0.09 mmol), and N,N-diisopropylethylamine (0.031 mL, 0.18 mmol). The mixture was subjected to conditions described in method C and purified by flash column chromatography (20% ethyl acetate/hexanes) to obtain 0.016 g (36%) of the title compound as a pink colored solid. [1]H NMR (400 MHz, CDCl$_3$) δ 8.52 (s, 1H), 8.09 (s, 1H), 7.74 (S, 1H), 7.62 (s, 1H), 7.58 (s, 1H), 7.39 (dd, J = 8.3, 1.9 Hz, 1H), 6.98 (d, J = 3.4 Hz, 1H), 6.90 (s, 1H), 6.87 (d, J = 8.0 Hz, 1H), 6.56–6.68 (m, 1H), 5.34 (s, 2H); LCMS m/z 492 (MH+).

## 1-{[2,4-Bis(trifluoromethyl)phenyl]methyl}-4-nitro-1H-pyrazole

**Chemical structure 3.** 1-{[2,4-Bis(trifluoromethyl)phenyl]methyl}-4-nitro-1H-pyrazole.

To a solution of 4-nitro-1H-pyrazole (0.2 g, 1.8 mmol) in N,N-dimethylformamide (2.0 mL), were added potassium carbonate (0.489 g, 3.5 mmol) and 2,4-bis(trifluoromethyl)benzyl bromide (0.332 mL, 1.8 mmol). The mixture was subjected to conditions described in method A and purified by flash column chromatography (15% ethyl acetate/hexanes) to obtain 0.43 g (72%) of the title compound as a white solid. [1]H NMR (400 MHz, CDCl$_3$) δ 8.15 (d, J = 8.1 Hz, 2H), 7.98 (s, 1H), 7.82 (d, J = 8.1 Hz, 1H), 7.39 (d, J = 8.1 Hz, 1H), 5.57 (s, 2H).

## N-(1-{[2,4-bis(trifluoromethyl)phenyl]methyl}-1H-pyrazol-4-yl)-5-(furan-2-yl)-1,2-oxazole-3-carboxamide (Ceapin-A7)

**Chemical structure 4.** N-(1-{[2,4-bis(trifluoromethyl)phenyl]methyl}-1H-pyrazol-4-yl)-5-(furan-2-yl)-1,2-oxazole-3-carboxamide (Ceapin-A7).

A solution of 1-{[2,4-bis(trifluoromethyl)phenyl]methyl}-4-nitro-1H-pyrazole (0.04 g, 0.1 mmol) in methanol (20 mL) was subjected to hydrogenation conditions described in general method B to obtain about 37 mg of crude 1-{[2,4-bis(trifluoromethyl)phenyl]methyl}-1H-pyrazol-4-amine as a colorless oil which was used without further purification. LCMS m/z 310 (MH+).

To a solution of 5-(furan-2-yl)-1,2-oxazole-3-carboxylic acid (0.018 g, 0.1 mmol) in N,N-dimethylformamide (0.5 mL), were added HATU (0.042 g, 0.11 mmol), 1-{[2,4-bis(trifluoromethyl)phenyl]methyl}-1H-pyrazol-4-amine (0.031 g, 0.1 mmol), and N,N-diisopropylethylamine (0.035 mL, 0.2 mmol). The mixture was subjected to conditions described in general method C to obtain 0.045 g (95%) of the title compound as a white powder. [1]H NMR (400 MHz, CDCl$_3$) δ 8.52 (s, 1H), 8.11 (s, 1H), 7.94 (s, 1H), 7.72 (d, J = 8 Hz, 1H), 7.67 (s, 1H), 7.58 (s, 1H), 7.09 (d, J = 8.1 Hz, 1H), 6.98 (d, J = 3.5 Hz, 1H), 6.90 (s, 1H), 6.56–6.57 (m, 1H), 5.58 (s, 2H); LCMS m/z 471 (MH+).

# 1-{[4-(Trifluoromethyl)phenyl]methyl}-4-nitro-1H-pyrazole

**Chemical structure 5.** 1-{[4-(Trifluoromethyl)phenyl]methyl}-4-nitro-1H-pyrazole.

To a solution of 4-nitro-1H-pyrazole (0.25 g, 2.21 mmol) in N,N-dimethylformamide (2.5 mL), were added potassium carbonate (0.61 g, 4.42 mmol) and 4-(trifluoromethyl)benzyl bromide (0.34 mL, 2.21 mmol). The mixture was stirred at ambient temperature for 18 hr and then subjected to conditions described in method A to obtain 0.61 g of the crude product as a white solid which was used without further purification. [1]H NMR (400 MHz, CDCl$_3$) δ 8.11 (d, $J$ = 2.7 Hz, 2H), 7.66 (d, $J$ = 8.0 Hz, 2H), 7.39 (d, $J$ = 8.0 Hz, 2H), 5.37 (s, 2H).

# N-(1-{[4-(Trifluoromethyl)phenyl]methyl}-1H-pyrazol-4-yl)-5-(furan-2-yl)-1,2-oxazole-3-carboxamide (Ceapin-A8)

**Chemical structure 6.** N-(1-{[4-(Trifluoromethyl)phenyl]methyl}-1H-pyrazol-4-yl)-5-(furan-2-yl)-1,2-oxazole-3-carboxamide (Ceapin-A8).

A solution of 1-{[4-(trifluoromethyl)phenyl]methyl}-4-nitro-1H-pyrazole (0.029 g, 0.1 mmol) in methanol (15 mL) was subjected to hydrogenation conditions described in method B to obtain about 0.026 g of crude 1-{[4-(trifluoromethyl)phenyl]methyl}-1H-pyrazol-4-amine as a colorless oil which was used without further purification

To a solution of 5-(furan-2-yl)-1,2-oxazole-3-carboxylic acid (0.019 g, 0.1 mmol) in N,N-dimethylformamide (0.5 mL), were added HATU (0.042 g, 0.11 mmol), 1-{[4-(trifluoromethyl)phenyl]methyl}-1H-pyrazol-4-amine (0.026 g, 0.1 mmol), and N,N-diisopropylethylamine (0.035 mL, 0.2 mmol). The mixture was subjected to conditions described in method C and purified by flash column chromatography (30% ethyl acetate/hexanes) to obtain 0.017 g (43%) of the title compound as a white powder. [1]H NMR (400 MHz, CDCl$_3$) δ 8.50 (s, 1H), 8.07 (s, 1H), 7.58–7.60 (m, 3H), 7.33 (d, $J$ = 7.9 Hz, 2H), 6.97 (d, $J$ = 3.5 Hz, 1H), 6.90 (s, 1H), 6.56 (dd, $J$ = 3.5, 1.7 Hz, 1H), 5.34 (s, 2H); LCMS $m/z$ 403 (MH+).

# 1-{[2-(Trifluoromethyl)phenyl]methyl}-4-nitro-1H-pyrazole

**Chemical structure 7.** 1-{[2-(Trifluoromethyl)phenyl]methyl}-4-nitro-1H-pyrazole.

To a solution of 4-nitro-1H-pyrazole (0.25 g, 2.21 mmol) in N,N-dimethylformamide (2.5 mL), were added potassium carbonate (0.61 g, 4.42 mmol) and 2-(trifluoromethyl)benzyl bromide (0.53 g, 2.21 mmol). The mixture was stirred at ambient temperature for 18 hr and then subjected to conditions described in method A to obtain 0.55 g of the crude 1-{[2-(trifluoromethyl)phenyl]methyl}-4-nitro-1H-pyrazole as a white solid which was used without further purification. $^1$H NMR (400 MHz, CDCl$_3$) δ 8.07–8.10 (m, 2H), 7.74 (d, J = 7.7 Hz, 1H), 7.48–7.59 (m, 2H), 7.28 (d, J = 8.0 Hz, 1H), 5.52 (s, 2H). LCMS m/z 272 (MH+).

## N-(1-{[2-(Trifluoromethyl)phenyl]methyl}-1H-pyrazol-4-yl)-5-(furan-2-yl)-1,2-oxazole-3-carboxamide (Ceapin-A9)

**Chemical structure 8.** N-(1-{[2-(Trifluoromethyl)phenyl]methyl}-1H-pyrazol-4-yl)-5-(furan-2-yl)-1,2-oxazole-3-carboxamide (Ceapin-A9).

A solution of 1-{[2-(trifluoromethyl)phenyl]methyl}-4-nitro-1H-pyrazole (0.032 g, 0.1 mmol) in methanol (15 mL) was subjected to hydrogenation conditions described in method B to obtain about 0.029 g of crude 1-{[2-(trifluoromethyl)phenyl]methyl}-1H-pyrazol-4-amine as a colorless oil which was used without further purification. LCMS m/z 242 (MH+).

To a solution of 5-(furan-2-yl)-1,2-oxazole-3-carboxylic acid (0.019 g, 0.1 mmol) in N,N-dimethylformamide (0.5 mL), were added HATU (0.042 g, 0.11 mmol), 1-{[2-(trifluoromethyl)phenyl]methyl}-1H-pyrazol-4-amine (0.026 g, 0.1 mmol), and N,N-diisopropylethylamine (0.035 mL, 0.2 mmol). The mixture was subjected to conditions described in method C and purified by flash column chromatography (25% ethyl acetate/hexanes) to obtain 0.028 g (70%) of the title compound as a white powder. $^1$H NMR (400 MHz, CDCl$_3$) δ 8.50 (s, 1H), 8.04 (s, 1H), 7.68 (d, J = 8.0 Hz, 1H), 7.66 (s, 1H), 7.58 (s, 1H), 7.45–7.49 (m, 1H), 7.40 (d, J = 7.6 Hz, 1H), 7.01 (d, J = 8.0 Hz, 1H), 6.97–6.99 (m, 1H), 6.90 (s, 1H), 6.55–6.57 (m, 1H), 5.52 (s, 2H); LCMS m/z 403 (MH+).

## Cell viability assay

U2-OS cells were detached using Accutase (Innovative Cell Technologies #AT-104-500) and the cell number was determined with the Scepter Cell Counter as described by the manufacturer (Millipore). 2000 cells were seeded into each well of a 96 well black walled optical bottom plate (Corning 3882) in normal growth medium. After 16–18 hr, the media was removed from the cells and 120 μL of fresh media with increasing concentrations of thapsigargin plus or minus 6 uM Ceapin 2 or 3. The highest concentration of thapsigargin was 90 nM and six 1:3 serial dilutions were performed. The final DMSO concentration for all samples including the DMSO only control was 0.034%. Breathable seals were used to seal the plate and placed in the incubator in a humidified chamber. At 72 hr, PrestoBlue Cell Viability Reagent (Life Technologies #A13262) was added to each well and incubated at 37C for 10 min and read on a plate reader as described by the manufacturer. PrestoBlue Cell Viability Reagent is a Resazurin based cell viability indicator that is reduced to a highly fluorescent molecule by viable cells. For data analysis the background (wells with media without cells) was subtracted from the experimental wells and viability relative to the DMSO treated wells was calculated in Microsoft Excel. The means of four independent experiments performed in triplicate were graphed using GraphPad Prism using the non-linear regression Sigmoidal, 4PL, X is log (Concentration) equation. The absolute EC50 was calculated in GraphPad Prism to interpolate X at 50% with 95% confidence intervals.

## Cell death assay

US-OS cells were detached with Accutase (Innovative Cell Technologies #AT-104-500) and cells counted using the Sceptor Cell Counter as described by the manufacture (Millipore). 20,000 cells in 0.5 mL of growth medium were added to each well of a 24 well plate (Corning 3526). After 16–18 hr the media was carefully removed from the wells and fresh medium with DMSO, 10, 30 or 90 nM Thapsigargin plus or minus 6 µM Ceapin 3 was added. Final DMSO concentration for all wells was 0.034%. After 72 hr, media was removed from each wells and placed in 1.5 mL Eppendorf tubes. 250 µL of Accutase was used to detach cells in the well and the entire volume was added to Eppendorf tube containing the kept media. 100 µL of the media and cell mixture in the Eppendorf tubes were transferred to 2 mL microtubes (VWR #16466–030) containing 100 µL of room temperature Muse Annexin V and Dead Cell Reagent (EMD Millipore #MCH100105). The kit contains Annexin V to detect phosphatidylserine for use as an early apoptotic marker and the membrane impermeable DNA dye, 7-Aminoactinomycin D (7AAD), to detect late apoptotic and necrotic cells. Tubes were gently vortexed for 5 s and incubated at room temperature for 20 min in the dark. Flow cytometry was performed using the Muse Cell Analyzer (EMD Millipore) and gated as directed by the manufacturer. To minimize reading times between samples, 1000 events were read for each sample and the percentage of live and apoptotic/dead cells was calculated as described by the manufacturer. The means of three independent experiments performed in duplicate were graphed in GraphPad Prism.

## Abbreviations used

7AAD: 7-aminoactinomycin D
ATF: Activating Transcription Factor
CFTR: Cystic Fibrosis transmembrane conductance regulator
CHOP: CCAAT-enhancer binding proteins (C/EBP) Homologous Protein
CMV: Cytomegalovirus
ERSE: ER Stress response Element
GFP: Green Fluorescent Protein
GRP: Glucose Regulated Protein
HEK293T: Human embryonic Kidney 293T cells.
IRE1: Inositol Requiring Enzyme One
MCP: Minimal cytomegaloviral promoter
PERK: Protein Kinase R (PKR)-like Endoplasmic Reticulum Kinase
RIDD: Regulated IRE1-Dependent Decay
Tg: Thapsigargin
Tm: Tunicamycin
U2-OS: U2 Osteosarcoma cells
UPRE: Unfolded Protein Response Element
VEGF: Vascular Endothelial Growth Factor
XBP1: X-box Binding Protein One

## Acknowledgements

The authors thank Carmela Sidrauski and Diego Acosta-Alvear for valuable input on high-throughput screening of UPR pathways and Samantha Zeitlin for input on high-content screening. We also thank Margaret Elvekrog, Mable Lam, Sandra Torres and Britt Adamson for critical reading of the manuscript and the members of the Walter lab for discussion and support. We also thank Pablo Valenzuela for his strong support. The S1P inhibitor Pf-429242 was a generous gift from Pfizer. pDEST-FRT-TO was a kind gift from Andrew N. Krutchinsky. Antibodies against ATF6α and ATF6β were kind gifts from Kazutoshi Mori and Thomas Rutkowski.

# Additional information

## Funding

| Funder | Grant reference number | Author |
| --- | --- | --- |
| Howard Hughes Medical Institute | Collaborative Innovation Award | Ciara M Gallagher<br>Carolina Garri<br>Erica L Cain<br>Kenny Kean-Hooi Ang<br>Christopher G Wilson<br>Steven Chen<br>Brian R Hearn<br>Priyadarshini Jaishankar<br>Michelle R Arkin<br>Adam R Renslo<br>Peter Walter |
| European Molecular Biology Organization | Long Term Fellowship | Ciara M Gallagher |
| National Cancer Institute | F32CA1777128 | Erica L Cain |
| Fundacion Ciencia Para La Vida | Project Basal PFB-16 and CONICYT | Carolina Garri |
| QB3 Malaysia Program | | Kenny Kean-Hooi Ang |
| Howard Hughes Medical Institute | | Peter Walter |

The funders had no role in study design, data collection and interpretation, or the decision to submit the work for publication.

## Author contributions

CMG, Conception and design, Acquisition of data, Analysis and interpretation of data, Drafting or revising the article; CG, High-throughput screening and qPCR (data acquisition), Analysis and interpretation of data, Editing the manuscript, Acquisition of data; ELC, Viability assays (design), Data acquisition, Analysis and interpretation of data, Writing the methods for viability assays, Editing the manuscript, Drafting or revising the article; KK-HA, CGW, High-throughput screening data acquisition, Analysis and interpretation of data, Acquisition of data; SC, High-content screening data acquisition, Acquisition of data; BRH, Chemistry design (scaffold analysis for secondary screen hits), Initial SAR of Ceapins, Acquisition of data, Analysis and interpretation of data; PJ, Chemistry design (SAR of Ceapins), Writing chemistry methods, Acquisition of data, Analysis and interpretation of data, Drafting or revising the article; AA-D, High-content screening (data analysis), Writing the methods for HCS data analysis, Analysis and interpretation of data, Drafting or revising the article; MRA, High-throughput screening data analysis and interpretation, Editing the manuscript, Analysis and interpretation of data, Drafting or revising the article; ARR, Chemistry design (SAR of Ceapins), Writing the manuscript, Analysis and interpretation of data, Drafting or revising the article; PW, Conception and design, Drafting or revising the article

## Author ORCIDs

Peter Walter, http://orcid.org/0000-0002-6849-708X

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
