## [Decision Letter]

Thank you for submitting your work entitled "Ceapins are a new class of unfolded protein response inhibitors, selectively targeting the ATF6α branch" for consideration by *eLife*. Your article has been favorably evaluated by Randy Schekman as the Senior editor and three reviewers: Peter Espenshade, Jeffery Kelly, and Davis Ng, who is a member of our Board of Reviewing Editors.

The reviewers have discussed the reviews with one another and the Reviewing Editor has drafted this decision to help you prepare a revised submission.

Summary:

The authors describe a new class of small molecule inhibitors specific for the ATF6α arm of the unfolded protein response. Using a cell-based reporter screen, just over 100K compounds were screened with extensive secondary screens to yield a single compound designated Ceapin-A1 that specifically inhibits ATF6α activation without effect on other arms of the UPR nor interfering with basic mechanisms involved in ATF6α processing. The molecule was altered further to yield several variants including inactive Ceapin-A5 and a higher potency Ceapin-A7. This is the first time the ATF6α pathway has been successfully "drugged" and completes the range of inhibitors for all three arms of the mammalian UPR. Interestingly, the authors also shed new light on the SKI-1/S1P inhibitor pf-429242's effect on ATF6 processing, which was previously not known in detail as it was used primarily to study its effects on viral membrane proteins.

Minor points:

1) Introduction, second paragraph: "principle" should be "principal"; "co-operatively" should be corrected.

2) Figure 1. Some images appear out of focus.

3) Figure 2. There appears to be far less GFP-ATF6 signal in panels C and D compared to A and B. Do the inhibitors affect steady state levels?

4) Figure 2. Consider changing one of the colors in the plot. It's difficult to discern dark grey and black.

5) Figure 2. It is unclear what is meant in the Y-axis label "relative to unstressed controls". Isn't the unstressed control represented by the first bar, at under 20%?

6) In the second paragraph of the subsection “Isolation of small molecule inhibitors of ER stress-induced nuclear translocation of GFP-ATF6α”: The distributions showed a wide range of responses within the population of cells in each well. Does this apply to Ceapin-A1? If so, please indicate the range. In other words, are a fraction of cells entirely unresponsive?

7) Figure 3—figure supplement 1: The authors should comment on why the compounds that passed through multiple filters should act so poorly on activation of endogenous genes.

8) Figure 3—figure supplement 4: While the mobility of ATF6α-M's mobility is obvious in Figure 3, it is not clear in this panel. Although I don't feel it necessary to re-do the experiment (given the other supporting data), either a better gel system or re-ordering of the lanes (ATF6α-N and ATF6α-M in adjacent lanes) to better show the difference is recommended in future experiments.

9) Figure 5. Ceapin-A5 appears to augment induction of BiP under both forms of stress. Is that correct?

10) Figure 5—figure supplement 1 is missing.

11) Ceapin does not inhibit ATF6β processing, but the positive control of S1P inhibitor has little effect. Presumably this is due to the issues discussed around ATF6-M, but these data are not convincing as presented since the positive control does not work well. A parallel experiment with cell fractionation as in Figure 3 would be more convincing.

12) Abstract, the phrase "detrimental in cancer models" is confusing. Perhaps rephrasing as "Activation of ATF6 promotes cell survival in cancer models."

13) In the second sentence of the subsection “Quantitative PCR analysis of mRNA expression to measure ATF6α target gene induction”, remove "at".

14) Figure 2/F would be improved if gray was lighter for "unstressed cells" to better differentiate from "stressed".

15) Shouldn't A5 be described as "the inactive ceapin analog A5" as "inactive ceapin" is confusing.

16) I assume the HEK293T cells used for screening are a stable cell line, if so please indicate in the text.

17) It would be a plus if Figure 2 were increased in size and much better described in the text.

---

## [Author Response]

*Minor points:*

*1) Introduction, second paragraph: "principle" should be "principal"; "co-operatively" should be corrected.*

Done.

*2) Figure 1. Some images appear out of focus.*

These images were acquired at low magnification using a high-throughput microscope. The panels in the original submission were cropped from these images and enlarged for the figure, leading to loss of resolution. We have replaced these panels with a crop of a larger area that clearly illustrates the point without the loss in resolution.

*3) Figure 2. There appears to be far less GFP-ATF6 signal in panels C and D compared to A and B. Do the inhibitors affect steady state levels?*

These images were acquired using a high-throughput microscope engineered to record qualitative differences and as such may be deceiving. In particular, the instrument performs automatic exposure adjustments, which explains the apparent discrepancies. For this reason, images were normalized internally prior to comparison between wells by calculating the ratio of nuclear to ER GFP signal (nuc:ER ratio) for each cell.

*4) Figure 2. Consider changing one of the colors in the plot. It's difficult to discern dark grey and black.*

Thank you for pointing this out. Unstressed has been changed from grey solid line to grey dashed line in the histogram and a grey-checkered box in the bar graph.

*5) Figure 2. It is unclear what is meant in the Y-axis label "relative to unstressed controls". Isn't the unstressed control represented by the first bar, at under 20%?*

We agree the axis labeling was confusing and have relabeled it “Percent activated cells”.

*6) In the second paragraph of the subsection “Isolation of small molecule inhibitors of ER stress-induced nuclear translocation of GFP-ATF6 α”: The distributions showed a wide range of responses within the population of cells in each well. Does this apply to Ceapin-A1? If so, please indicate the range. In other words, are a fraction of cells entirely unresponsive?*

We have now included these data as Figure 2—figure supplement 3. The range for Ceapin-A1 is similar to the other treatments – the difference between treatments is not the range per se but the position of the histogram on the X-axis. In the secondary screen, a fraction of the cells are indeed unresponsive to Ceapin-A1 (see where the threshold crosses the histogram for Ceapin-A1). This concentration of this aliquot of Ceapin-A1 inhibits nuclear translocation of GFP-ATF6 to 35.07 ± 18.15%.

This low activity is because the concentration of active Ceapin-A1 tested in the secondary screen is below its IC_50_. Ceapin-A1 in the library is less active than repurchased stocks (IC_50_ of 8.49 μM compared to 4.9 ± 1.2 μM in ERSE-luciferase assays). Additionally, for the secondary screen all compounds were tested at a single concentration of 6.6 μM, which is below the IC_50_ of the library aliquot of Ceapin-A1. For this reason, these data were not included in the original version of the manuscript as they report on the potency of Ceapin-A1 in the library stock and not the potency of fresh stocks.

*7) Figure 3—figure supplement 1: The authors should comment on why the compounds that passed through multiple filters should act so poorly on activation of endogenous genes.*

We do not know the reason for this discrepancy. We exclusively focused on the Ceapin series after obtaining the results in Figure 3—figure supplement 1, which validated the effects of the compound on endogenous gene expression and additionally had an increased stringency filter applied. We have added this explanation to the figure legend of Figure 3—figure supplement 1.

*8) Figure 3—figure supplement 4: While the mobility of ATF6α-M's mobility is obvious in Figure 3, it is not clear in this panel. Although I don't feel it necessary to re-do the experiment (given the other supporting data), either a better gel system or re-ordering of the lanes (ATF6α-N and ATF6α-M in adjacent lanes) to better show the difference is recommended in future experiments.*

Thank you, point taken.

9) Figure 5. Ceapin-A5 appears to augment induction of BiP under both forms of stress. Is that correct?

No. We have done this experiment three times each with two different antibodies and augmented induction of BiP by the inactive Ceapin analog A5 was only observed in this one particular blot. We have replaced the figure with data from a representative experiment.

10) Figure 5—figure supplement 1 is missing.

We have included the missing supplement in the revised manuscript.

11) Ceapin does not inhibit ATF6β processing, but the positive control of S1P inhibitor has little effect. Presumably this is due to the issues discussed around ATF6-M, but these data are not convincing as presented since the positive control does not work well. A parallel experiment with cell fractionation as in Figure 3 would be more convincing.

We have added new data to address this question. We performed differential centrifugation on U2-OS cells and looked for endogenous ATF6α and ATF6β. We detected both ATF6α-N and ATF6β-N in the nuclear extract of cells subjected to ER stress.

Although both ATF6α and ATF6β were cleaved in ER stressed cells treated with the site-1 protease inhibitor, analogous to the full-length proteins, neither cleavage product was detected in the nuclear fraction, indicating that neither ATF6α-N nor ATF6β-N were generated. These data suggest that the cleavage product detected by the anti-ATF6β antibody is ATF6β-M.

In ER stressed cells treated with Ceapin-A7 we reproducibly found that while ATF6α is not cleaved and there is no ATF6α-N in the nuclear fraction, ATF6β is cleaved and ATF6β-N is present in the nuclear fraction. These data strengthen the argument that Ceapins specifically inhibit processing of ATF6α but not ATF6β.

12) Abstract, the phrase "detrimental in cancer models" is confusing. Perhaps rephrasing as "Activation of ATF6 promotes cell survival in cancer models."

Thank you for the suggestion – we changed it.

13) In the second sentence of the subsection “Quantitative PCR analysis of mRNA expression to measure ATF6α target gene induction”, remove "at".

Done.

14) Figure 2/F would be improved if gray was lighter for "unstressed cells" to better differentiate from "stressed".

Done – see point 4 above.

15) Shouldn't A5 be described as "the inactive ceapin analog A5" as "inactive ceapin" is confusing.

We agree and have changed this call-out in both manuscripts.

16) I assume the HEK293T cells used for screening are a stable cell line, if so please indicate in the text.

This fact was already included in the methods. We have now added a statement to the Results section also.

17) It would be a plus if Figure 2 were increased in size and much better described in the text.

We have increased Figure 2 in size. To facilitate this change and keep Figure 2 on a single page panel G from the original figure has been moved to Figure 2—figure supplement 2. We have edited the text to clarify why this assay was performed and the results interpreted from the images.